# Battle of the Backbones: A Large-Scale Comparison of Pretrained Models across Computer Vision Tasks

**Micah Goldblum**[1] *    **Hossein Souri**[2] *    **Renkun Ni**[3]    **Manli Shu**[3]    **Viraj Prabhu**[4]

**Gowthami Somepalli**[3]    **Prithvijit Chattopadhyay**[4]    **Mark Ibrahim**[6]    **Adrien Bardes**[5,6]

**Judy Hoffman**[4]    **Rama Chellappa**[2]    **Andrew Gordon Wilson**[1]    **Tom Goldstein**[3]

## Abstract

Neural network based computer vision systems are typically built on a *backbone*, a pretrained or randomly initialized feature extractor. Several years ago, the default option was an ImageNet-trained convolutional neural network. However, the recent past has seen the emergence of countless backbones pretrained using various algorithms and datasets. While this abundance of choice has led to performance increases for a range of systems, it is difficult for practitioners to make informed decisions about which backbone to choose. Battle of the Backbones (BoB) makes this choice easier by benchmarking a diverse suite of pretrained models, including vision-language models, those trained via self-supervised learning, and the Stable Diffusion backbone, across a diverse set of computer vision tasks ranging from classification to object detection to OOD generalization and more. Furthermore, BoB sheds light on promising directions for the research community to advance computer vision by illuminating strengths and weakness of existing approaches through a comprehensive analysis conducted on more than 1500 training runs. While vision transformers (ViTs) and self-supervised learning (SSL) are increasingly popular, we find that convolutional neural networks pretrained in a supervised fashion on large training sets still perform best on most tasks among the models we consider. Moreover, in apples-to-apples comparisons on the same architectures and similarly sized pretraining datasets, we find that SSL backbones are highly competitive, indicating that future works should perform SSL pretraining with advanced architectures and larger pretraining datasets. We release the raw results of our experiments along with code that allows researchers to put their own backbones through the gauntlet here: https://github.com/hsouri/Battle-of-the-Backbones.

## 1   Introduction

The dominant paradigm for building machine vision systems involves a feature extractor network, also known as a *backbone*, which feeds into a task-specific *head*. The backbone might output a dense array of features for object detection and localization, or a single feature vector for classification or image retrieval. While backbones can be trained from scratch on task-specific data, many off-the-shelf backbones are pretrained on large benchmark datasets and then fine-tuned for the task at hand. This transfer learning approach has several advantages. First, it dramatically reduces the application-specific data requirements of deep learning and has led to improved performance on a wide range of

---

*Authors contributed equally. Correspondence to goldblum@nyu.edu and hsouri1@jhu.edu. This work was conducted at New York University[1], Johns Hopkins University[2], University of Maryland[3], Georgia Institute of Technology[4], Inria[5], and Meta AI Research[6].

applications. Second, it can speed up training and reduce compute costs even when large amounts of task-specific data are available [29]. Finally, pretraining datasets often contain images from many disparate domains, resulting in model robustness that can be transferred to downstream tasks.

Early deep learning based vision systems relied heavily on ImageNet pretraining [23, 59]. In contrast, today's practitioners have access to a cornucopia of choices, with different pretrained models resulting in significant performance differences. There are three primary factors that influence the performance of such a model: its architecture, the pretraining algorithm, and the pretraining dataset. Each of these design dimensions presents many options, resulting in a dizzying array of choices for practitioners building a computer vision system. Despite this wide variety of choices, practitioners have no resource to turn to and instead are left piecing together results from method papers or testing out the backbones themselves.

We pit these backbones against each other in a *Battle of the Backbones* (BoB). BoB compares many popular publicly available pretrained checkpoints, as well as randomly initialized baselines, on a wide variety of downstream tasks including image classification on natural, medical, and satellite images (Section 3.1), object detection and segmentation (Section 3.2), out-of-distribution generalization (Section 3.3), and image retrieval (Section 3.4).

Aside from assisting practitioners building computer vision systems, another central goal of this benchmark is to help guide the research community towards fruitful research directions in their quest for designing better backbones. BoB sheds light on the strengths and weaknesses of pretraining routines and architectures, revealing popular misconceptions and fundamental limitations, as well as promising directions for improvement. Below, we summarize several of our primary findings and discuss previous efforts for comparing backbones.

## 1.1 Battle of the Backbones: The TLDR

The subsequent sections in this paper contain numerous experimental details. Therefore, we distill several key findings below:

▷ Across the suite of comprehensive evaluations in BoB, spanning tasks, datasets, and settings (including ID and OOD), supervised ConvNeXt-Base, supervised SwinV2-Base trained using ImageNet-21k, and CLIP ViT-Base come out on top. The same winners also win at smaller scales. Among smaller backbones, ConvNeXt-Tiny and SwinV2-Tiny emerge victorious, followed by DINO ViT-Small.

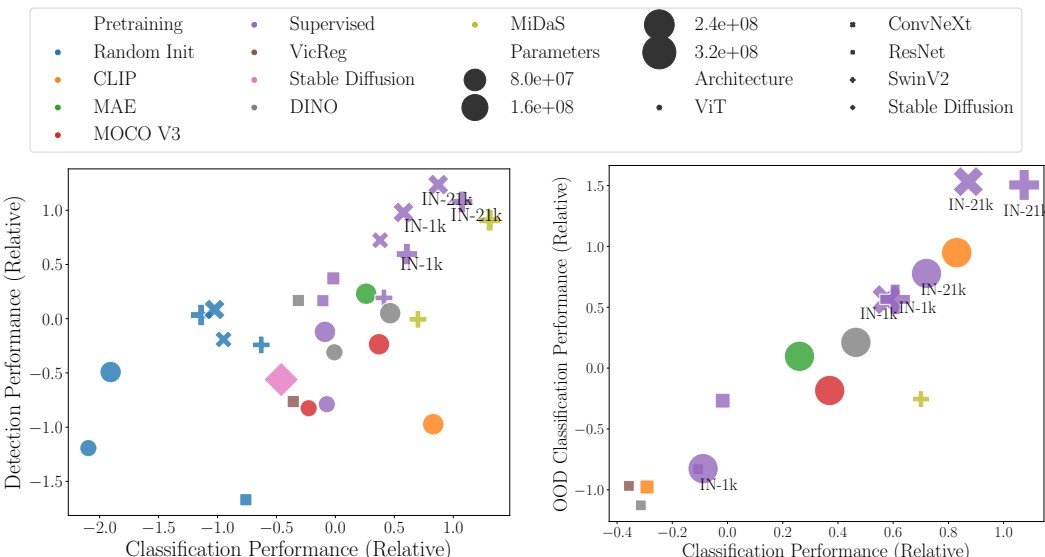

Figure 1: **Performance is correlated across tasks.** Performance for each model is reported in terms of standard deviations above/below the mean averages across datasets. **Left:** Comparison between classification and detection. **Right:** Comparison between classification and OOD classification.

▷ Despite the recent attention paid to transformer-based architectures and self-supervised learning, high-performance convolutional networks pretrained via supervised learning outperform transformers on the majority of tasks we consider.

▷ The observed superiority of supervised pretraining occurs because such models are often trained on larger datasets. In apples-to-apples comparisons on the same dataset scale, SSL models outperform their supervised counterparts.

▷ ViTs are more sensitive to the amount of pretraining data and the number of parameters than CNNs.

▷ Performance across tasks is strongly correlated – the top-performing backbones in BoB tend to be universally good across tasks and settings. See Figure 1.

## 1.2 Previous Benchmarks

Throughout much of the last decade, the most popular backbones were pretrained on ImageNet [17]. Since 2020, SimCLR [10] and CLIP [73] have popularized self-supervised backbones and spawned much new research. While method papers that propose a new pretraining routine typically compare to similar competitors on several downstream tasks, we focus in this section on works that specifically benchmark large collections of backbones on diverse tasks.

In 2019, Goyal et al. [25] compared AlexNet [47] and ResNet-50 [28] models pretrained using `colorization` and `jigsaw` pretext tasks to supervised learning models, finding that supervised learning massively outperformed SSL at the time. Kolesnikov et al. [44] similarly compared several pretext tasks and convolutional neural network architectures, showing that architectural advances on supervised learning do not always translate to improved self-supervised learning. Kornblith et al. [45] instead benchmarked the transferability of ImageNet-trained supervised learning models on downstream classification tasks, varying the architecture and finding that the correlation between downstream performance and ImageNet test accuracy is nearly perfect across architectures. In the same year, Zhai et al. [107] built the Visual Task Adaptation Benchmark (VTAB) and tested various self-supervised learning methods including VAEs and GAN discriminators, also exhibiting the dominant performance of supervised learning models. In 2020, Ericsson et al. [21] evaluated ResNet-50 models trained on ImageNet using various SSL algorithms, finding that the performance of then-existing SSL algorithms on a richer set of downstream tasks were strongly correlated with their ImageNet-1k test accuracy and finding improved performance of the newer SSL algorithms compared to previous studies.

Since the above works, pretraining algorithms along with their training sets and architectures have made tremendous progress, and whereas supervised learning was previously the default approach to pretraining, the options now are endless. Therefore, benchmarking backbones deserves renewed attention. See Appendix A for an additional survey of task-specific benchmarks.

## 2 A Guide to BoB

Among the distinguishing features of the diverse backbones competing in our battle are their architectures, pretraining routines, and the datasets on which they were pretrained. Table 1 contains an overview of the backbones we benchmark including their pretraining algorithms, pretraining datasets, and architectures. We also provide a more detailed description of these features and the precise pretrained checkpoints we use in Appendix B.

**A Note on Scale and Apples-to-Apples Comparison.** *Many practitioners have limited compute and moreover will need to tune hyperparameters on their own datasets without exceeding their compute budget. To simulate this scenario, we perform moderate hyperparameter sweeps, we preclude particularly long training schedules, and we do not consider architectures bigger than ConvNeXt-Base, except for the Stable Diffusion backbone which does not come in a smaller size. Specific hyperparameter grids are detailed in subsequent sections. Moreover, we only use publicly available checkpoints that would also be accessible to practitioners. Available checkpoints were pretrained with varying amounts of hyperparameter tuning, and different pretraining algorithms were trained on different datasets and architectures making a precise apples-to-apples comparison infeasible. Nevertheless, this comparison of existing checkpoints is the relevant one for practitioners, as it represents realistic conditions, and we use identically sized hyperparameter sweeps for each backbone on downstream tasks.*

Table 1: **A synopsis of the backbones we benchmark.** Columns correspond to the pretraining algorithm, a coarse categorization, the pretraining dataset, and the architectures we include. A detailed description of each algorithm, pretraining dataset, and architecture can be found in Appendix B.

| Pretraining | Style | Dataset | Architecture(s) |
|---|---|---|---|
| MoCo v3 [12] | SSL | ImageNet-1k [17] | ViT [18] |
| VICReg [3] | SSL | ImageNet-1k | ResNet [28] |
| VICRegL [4] | SSL | ImageNet-21k | ConvNeXt [58] |
| DINO [8] | SSL | ImageNet-1k | ResNet, ViT |
| MAE [30] | SSL | ImageNet-1k | ViT |
| Stable Diffusion [77] | Vision-Language | LAION-2B [81] | Stable Diffusion encoder |
| CLIP [73] | Vision-Language | LAION-2B, CLIP | ResNet, ViT |
| MiDaS [75] | Supervised | 12 × Depth Datasets | SwinV2 [57] |
| Image classification | Supervised | ImageNet-21k,-1k | All above architectures |
| Random initialization | None | N/A | All above architectures |

## 2.1 The Tasks

In order to comprehensively probe the capabilities of the backbones, we evaluate their performance both fine-tuned and frozen on a number of downstream tasks belonging to the following categories:

- **Classification:** We measure both fine-tuned and linear probe performance of backbones on various downstream classification tasks including natural, medical, or satellite image datasets in Section 3.1. Image classification tasks require that a backbone extract features which identify the content of an image's foreground but not necessarily how many of an object there are or where they are located within an image.

- **Object detection and segmentation:** Unlike image classification, dense prediction tasks require backbones to extract features containing the precise locations of objects, on a pixel basis for segmentation and in enough fidelity to draw bounding boxes for object detection. We evaluate backbones on both of these tasks in Section 3.2.

- **Out-of-distribution generalization:** In real-world applications, computer vision systems are often deployed on data which does not reflect their training set distribution. Even high-performing models are known to fail under domain shifts [71, 32]. Therefore, we evaluate the abilities of models both to generalize to new downstream domains in Section 3.3.

- **Image retrieval:** Image retrieval requires a backbone to match like images via proximity in feature space. We explore tasks that require matching the images with respect to various criteria such as semantic content and visual similarity in Section 3.4.

## 3 Experimental Setup

We now describe our experimental setup for each task. Specifically, we list learning protocols, datasets, and evaluation metrics. Find complete experimental and implementation details in Appendix C.

## 3.1 Classification

**Learning protocols.** We evaluate pretrained backbones on various datasets under two fine-tuning protocols, following previous works [12, 30, 8, 10]: **end-to-end fine-tuning** (including experiments with only a small number of labeled samples) and **linear probing**. In the former scenario, we fine-tune the full model end-to-end on a given dataset or on a fraction of it, and we measure the accuracy on the test split. In the linear probing scenario, we extract features from the frozen pretrained backbone, and only learn a linear classifier on top of these pretrained representations. These two protocols are widely used in previous work to evaluate the quality of pretraining methods such as in self-supervised learning [12, 30, 8, 10] and vision-language pretraining [1, 106].

**Datasets and evaluation metrics.** We conduct experiments on 6 common image classification datasets, covering multiple domains such as natural images (ImageNet-1K [17], CIFAR-100 [46],

Flowers-102 [65], Aircraft [61]), satellite images (EuroSAT [31]), and medical X-ray data (CheXpert [37]) showing the generalization and transferability of the pretrained backbones. All datasets we use are publicly available, and we list their details including size and the number of classes in Appendix C. For experiments with only a fraction of the training set, we randomly sample 1% and 10% of the training samples and fine-tune the pretrained backbones on these subsets. When sampling the subsets, we maintain the original dataset's label distribution. Note that we only consider in-domain generalization here, where the training and testing splits are from the same source.

To evaluate, we measure *classification accuracy* and *Area Under the ROC Curve* (AUC) on the test split as performance metrics for single-label and muti-label classification tasks, respectively. In addition to the best score among hyperparameter vectors, we also plot the accuracy for the first several epochs to show the convergence rate of different pretrained backbones. Moreover, we benchmark the latency and the memory usage of each backbone on the same device.

## 3.2 Object Detection and Segmentation

**Learning protocols.** For evaluations on object detection and instance segmentation, we employ the Cascade Mask R-CNN framework [5]. We conduct experiments with three protocols: **(1)** end-to-end training from random initialization, **(2)** end-to-end finetuning using pretrained backbones, and **(3)** finetuning with frozen backbones. Whereas finetuning with a frozen backbone is atypical in segmentation and detection, this latter protocol allows us to probe localization within features extracted by pretrained models and complements linear probing classification experiments. See Appendix C.1 for a discussion on the potential for ViTs, especially large ones, to exceed the performance of other models under more expensive training protocols.

**Datasets and evaluation metrics.** We conduct object detection and instance segmentation evaluations on the popular COCO dataset [54]. We follow the COCO-style average precision (AP) metric, which calculates the average across various Intersection over Union (IoU) thresholds. We report the box Average Precision (box AP), box AP@50, and AP@75 for object detection and mask Average Precision (mask AP), mask AP@50, and mask AP@75 for instance segmentation [55].

## 3.3 Out-of-Distribution Generalization

While modern networks may exhibit strong performance on data distributions they are trained on, a wide body of prior work [71, 32] has found that the performance of such models can degrade significantly under distribution shifts. In addition to evaluating the in-distribution performance of backbones across a diverse set of downstream tasks, we also consider how this performance translates to out-of-distribution (OOD) settings.

**Learning protocols.** Several task-specific datasets and benchmarks have been proposed to evaluate the robustness of models to deviations from their training distributions. Concretely, we study the generalization of the trained backbones on two tasks, **(1)** image classification and **(2)** object detection, and on two types of distribution shifts, **(A)** structure and style variations within ImageNet and **(B)** synthetic-to-real generalization.

**Datasets and evaluation metrics.** We consider the following broad benchmarks for OOD evaluation:

**(A) Robustness to changes in structure and style.** We measure OOD generalization of ImageNet-trained or fine-tuned models on the following benchmarks: **(i)** ImageNet-A [34]. ImageNet-A(dversarial) contains a curated subset of ImageNet test images spanning 200 categories that are especially challenging for trained deep models. **(ii)** ImageNet-V2 [76]. ImageNet-V2 is an additional test set of ImageNet-like images collected a decade after the original dataset following an identical collection protocol. **(iii)** ImageNet-R [33]. ImageNet-R(endition) contains artistic renditions for 200 categories from ImageNet, including cartoons, graffiti, embroidery, origami, sculptures, *etc.* **(iv)** ImageNet-S [93]. ImageNet-S(ketch) is a web-crawled and manually cleaned collection of black and white sketch images from ImageNet categories.

**(B) Syn-to-real generalization.** We also measure the performance of models trained on synthetic data and tested on real data. Synthetic data has emerged as a popular alternative in settings where it may be hard or expensive to curate reliably annotated real-world data. We measure syn-to-real generalization for image classification and object detection on the two following popular benchmarks: **(i)** VisDA Syn→Real. The VisDA classification benchmark consists of $\sim$ 152k synthetic images and

$\sim$ 55k real images across 12 classes. The synthetic images in VisDA are 3D renderings of objects from multiple viewpoints and under different lighting conditions. The real counterparts are crops of the 12 classes obtained from the COCO dataset. **(2)** Sim10k$\rightarrow$Cityscapes. For object detection, we use Sim10k as the synthetic training dataset and Cityscapes as the real evaluation dataset. Sim10k consists of $\sim$ 10k street view images (drawn from GTAV). Cityscapes consists of $\sim$ 5k densely annotated street view images curated from vehicular viewpoints in the real world. Following prior work [13], we train on the entirety of Sim10k to detect instances of "car" and measure detection performance on the validation split of Cityscapes.

We report generalization performance using classification accuracy on the OOD test set for image classification and mean average precision or mAP@50 for object detection.

### 3.4 Image Retrieval

We conduct evaluations on a diverse set of retrieval datasets encompassing content-based image retrieval and classification datasets that we repurpose for semantic retrieval tasks. For geographic landmark retrieval, we utilize the Oxford dataset [69] and the Paris dataset [70]. To ensure accuracy, we employ the cleaned-up versions of these datasets with corrected labels [72]. The INSTRE dataset [95] consists of objects such as toys and irregularly-shaped products placed in different locations and conditions. To examine fine-grained retrieval, we employ the Caltech-UCSD Birds-200 dataset (CUB-200) [91], which contains various bird classes captured under different backgrounds, poses, and lighting conditions. For a diverse set of natural images, we use the iNaturalist dataset [88]. This dataset offers a wide range of fine-grained categories classified into 13 super-categories, including Plant, Insect, Bird, and Mammal. To evaluate retrieval performance in real-world scenarios, we employ the Objectnet dataset [2]. This dataset consists of 313 object classes with randomly varying backgrounds, rotations, and imaging viewpoints. For large-scale landmark recognition, we utilize the Google Landmarks v2 dataset [99], which includes approximately 200,000 unique landmarks. Lastly, we employ the INRIA Copydays dataset [19], which comprises a small collection of holiday photos. Among the datasets mentioned, iNaturalist, Objectnet, and CUB-200 can be categorized as semantic retrieval datasets, while the remaining datasets fall under content-based retrieval datasets.

To evaluate, we measure model performance using mean-Average-Precision or *mAP* [68]. We first compute the average precision for a given query image, and then compute the mean over all queries to find the mAP. We also measure *Recall@k*, which measures the proportion of correct matches among the top $k$, and *MRR* (Mean Reciprocal Rank), which records the number of results returned before the first correct match and computes the mean of the reciprocal of these misses. Higher is better for all metrics.

## 4 I'm a Practitioner. Which Backbone Should I Choose?

Practitioners today can choose from a large catalogue of backbones of varying sizes, training methods, and pretraining data: which backbone should a practitioner select for a particular task or in general? To answer this question, in BoB, we systematically compare publicly available backbones (see Table 1) across multiple tasks, datasets and settings. To make these comparisons, we use the following ranking protocol:

**(1) Setting-specific Z-Scores.** For a particular task and setting (e.g, top-1 classification accuracy on ImageNet), we first compute z-scores for all the backbones being evaluated – i.e., for setting specific performance (e.g., accuracy) values $\{x_i\}_{i=1}^N$, z-scores are computed as $\{\frac{x_i-\mu}{\sigma}\}_{i=1}^N$ where $\mu$ and $\sigma$ are the mean and standard deviation of the sample. This allows us to measure how good a specific backbone is (stds above or below) compared to "mean" performance of all backbones in that setting.

**(2) Cross-setting Comparisons.** To compare backbones across different tasks and settings, we simply aggregate and compare the previously obtained z-scores to obtain a relatively (coarse) ranking of backbones.

Using rankings, we can report not only the best performing backbones for each task but also the best backbone in terms of overall performance across tasks, datasets and settings (see Table 2 for a summary).

Table 2: **Which backbone should I choose?** We list the top 3 most performant backbones (left to right) for various tasks and settings. Red corresponds to OOD evaluations and Green indicates overall comparisons.

| Task | Good | Better | Best |
|------|------|--------|------|
| 1 Cls | ConvNeXt-B (IN-21k) | CLIP ViT-B (LAION-2B) | Sup. SwinV2-B (IN-21k,1k) |
| 2 Det | Sup. ConvNeXt-B (IN-1k) | Sup. SwinV2-B (IN-21k,1k) | Sup. ConvNeXt-B (IN-21k) |
| 3 Seg | Sup. ConvNeXt-B (IN-1k) | Sup. SwinV2-B (IN-21k,1k) | Sup. ConvNeXt-B (IN-21k) |
| 4 Ret | CLIP ViT-B (LAION-2B) | Sup. SwinV2-B (IN-21k,1k) | Sup. ConvNeXt-B (IN-21k) |
| 5 (OOD) Cls | CLIP ViT-B (LAION-2B) | Sup. SwinV2-B (IN-21k,1k) | Sup. ConvNeXt-B (IN-21k) |
| 6 (OOD) Det | Sup. ConvNeXt-B (IN-21k) | Sup. ConvNeXt-T (IN-1k) | Sup. ConvNeXt-B (IN-1k) |
| 7 All | CLIP ViT-B (LAION-2B) | Sup. SwinV2-B (IN-21k,1k) | Sup. ConvNeXt-B (IN-21k) |

## 4.1 Task-Specific Backbones

**Classification.** For classification, across multiple datasets and experimental settings (fine-tuning, linear probing, full and low-shot training), we find "Supervised SwinV2-Base trained on IN-21k (finetuned on IN-1k)" to be the best performing backbone, followed by "CLIP ViT-Base" and "Supervised ConvNeXt-Base trained on IN-21k" (see row 1, Table 2).[2]

**Object Detection & Segmentation.** For object detection and instance segmentation, we find "Supervised ConvNeXt-Base trained on IN-21K" > "Supervised SwinV2-Base trained on IN-21k (finetuned on IN-1k)" > "Supervised ConvNeXt-Base trained on IN-1k".

**Image Retrieval.** For image retrieval, we find "Supervised ConvNeXt-Base trained on IN-21k" to be the best choice, with "Supervised SwinV2-Base trained on IN-21k (finetuned on IN-1k)" and "CLIP ViT-B trained on LAION-2B" being second and third.

**(OOD) Classification.** Across OOD evaluations for classification, we find "Supervised ConvNeXt-Base trained on IN-21k" > "Supervised SwinV2-B trained on IN-21k (finetuned on IN-1k)" > "CLIP ViT-Base trained on LAION-2B".

**(OOD) Object Detection.** For Syn→Real object detection, we find "Supervised ConvNeXt-Base trained on IN-1k" to be the best backbone, followed by "Supervised ConvNeXt-Tiny trained on IN-1k" and "Supervised ConvNeXt-Base trained on IN-21k".

## 4.2 Best Backbones Overall

For practitioners with no specific task in mind, the best performing models in terms of aggregate performance are "Supervised ConvNeXt-Base trained on IN-21k" followed by "Supervised SwinV2-Base trained on IN-21k (finetuned on IN-1k)" and "CLIP ViT-Base trained on LAION-2B". Overall, we note that backbones trained in a supervised fashion (SwinV2-Base, ConvNeXt-Base) or with vision and language supervision (CLIP ViT-Base) outperform the rest. Furthermore, we find that CLIP ViT-Base is closely followed by Supervised ViT-Base trained on IN-21k (finetuned on IN-1k). We more precisely compare approaches and analyze trends in Section 5.

## 4.3 Backbones on a Tight Budget

Many computer vision applications demand efficient backbones for fast or on-device inference. In this section, we benchmark three small backbones: RegNetX-400F [74], EfficientNet-B0 [84] and ResNet-18 [28] all pretrained in a supervised fashion on ImageNet-1k. We rank the performance of these small backbones on the set of tasks in Table 3. We find that EfficientNet-B0 performs best overall and across classification, retrieval, and OOD classification, followed by RegNetX-400MF and then ResNet-18. Interestingly, ResNets still outperform newer efficient architectures for detection and segmentation.

---

[2]To ensure fair comparisons across backbones, we exclude MiDaS variants evaluated on ImageNet for this comparison.

Table 3: **Which tiny backbone should I choose?** We rank the most performant very lightweight backbones (left to right) for various tasks and settings. Red correspond to OOD evaluations and Green indicates overall comparisons.

| Task | Good | Better | Best |
|------|------|--------|------|
| 1 Cls | ResNet-18 | RegNetX-400MF | EfficientNet-B0 |
| 2 Det | RegNetX-400MF | EfficientNet-B0 | ResNet-18 |
| 3 Seg | RegNetX-400MF | EfficientNet-B0 | ResNet-18 |
| 4 Ret | ResNet-18 | RegNetX-400MF | EfficientNet-B0 |
| 5 (OOD) Cls | ResNet-18 | RegNetX-400MF | EfficientNet-B0 |
| 6 (OOD) Det | EfficientNet-B0 | ResNet-18 | RegNetX-400MF |
| 7 All | ResNet-18 | RegNetX-400MF | EfficientNet-B0 |

## 5 Observations and Trends

▷ **A performance comparison of ViTs and CNNs. Modern architectures strongly outperform vanilla ViTs.** We see in Table 2 that the best performing backbone (ConvNeXt-Base) is convolutional, with a hierarchical transformer (SwinV2-Base) being a close second. The latter transformer architecture incorporates a strong spatial inductive bias. These findings suggest that the community should move past vanilla ViTs which are still used frequently. As a caveat, we do not evaluate very large models, and it is possible that ViTs might outperform their more advanced variants or convolutional networks at larger scales.

▷ **ViTs benefit more from scale than CNNs.** For the suite of backbones considered in BoB, we find that relative performance (z-scores) for both CNNs and ViTs correlates positively with parameter count but more so for ViTs (spearman $\rho = 0.58$) than for CNNs (spearman $\rho = 0.35$). Similarly, while overall relative performance correlates with the size of pretraining data, the correlation is again significantly higher for ViTs ($\rho = 0.72$) than for CNNs ($\rho = 0.33$). This observation indicates that benchmarking much larger backbones might yield different winners, possibly ones with transformer-based architectures.

▷ **Supervised or not? Supervised learning backbones dominate, but primarily because they are available pretrained on larger datasets. SSL backbones can outperform supervised pre-training with similar sized pre-training datasets.** We obtain the average score of the top 3 backbones within different pretraining styles, namely self-supervised, supervised with ImageNet-1K, and supervised with ImageNet-21K, for each task (see Appendix D). ConvNeXt and SwinV2 pretrained with supervision on ImageNet-21K outperform the SSL backbones on all tasks. The results suggest that we should try using advanced architectures, either convolutional or transformers, when applying SSL methods, and we should train on large datasets to compete with supervised learning. In these experiments, supervised pretraining checkpoints are often available trained on much larger datasets (ImageNet-21k). When comparing models pretrained on similarly sized datasets, SSL or vision-language pretraining methods achieve better performance on classification (both in- and out-of-distribution) and retrieval tasks, which heavily rely on the learned representations. However, supervised learning backbones maintain a decisive edge for detection and segmentation. We can also compare backbones which use the same ViT-Base architecture and find that SSL methods do outperform ImageNet-1k supervised backbones but are worse than ImageNet-21k trained backbones.

▷ **Performance across tasks is highly correlated.** Across tasks examined, we find a strong positive Spearman correlation between performance on task pairs (typically $\rho > 0.8$). This finding supports the current trend of general purpose foundation models for computer vision. Moreover, this finding also supports recent work which argues that a single inductive bias can solve a wide range of seemingly different problems [24]. However, it is noteworthy that the retrieval task exhibited a comparatively lower but still statistically significant correlation ($\rho = 0.49$) with respect to classification and retrieval ranking. This lower correlation can be attributed to the performance limitations of the MiDaS and MAE pretrained models in the context of retrieval. Upon removing these two backbones, the correlation coefficient $\rho$ increased to 0.8, reinforcing the influence of the aforementioned models on the observed results.

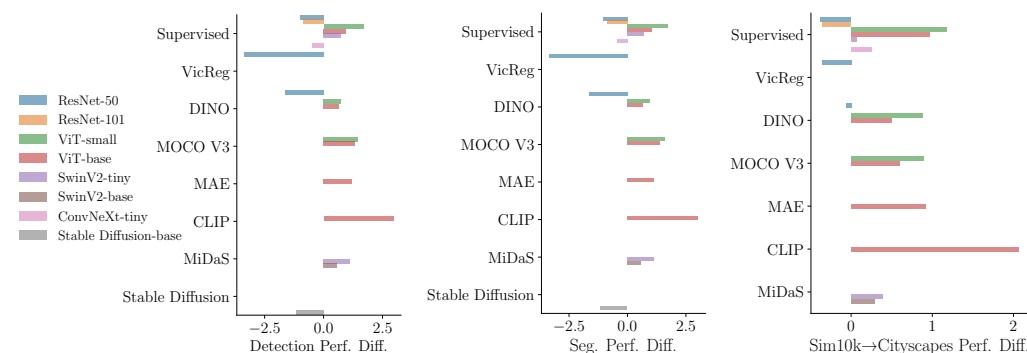

Figure 2: **Transformers benefit significantly more from end-to-end fine-tuning than CNNs on dense prediction tasks.** We visualize the difference in performance between end-to-end fine-tuning and only training the head atop a frozen feature extractor on different tasks. The x-axis is the difference in relative performance (fine-tuning z-score minus fixed backbone z-score). Across panels, the performance differences correlate between tasks.

▷ **Transformers excel under end-to-end fine-tuning while convolutional networks excel under linear probing.** For "linear probing" experiments, we freeze a pretrained backbone and only learn the head. Note that for detection and segmentation, the head is more than a linear layer. By inspecting the performance difference between the two fine-tuning strategies (Figure 2), we find that ViTs benefit significantly more from end-to-end fine-tuning compared to CNNs, both for supervised and self-supervised pretraining. See Figure 2 for a comparison on dense prediction tasks.

▷ **CLIP models and the promise of advanced architectures in vision-language modeling.** For almost all the tasks (except OOD detection), CLIP pretraining is the best among the vanilla vision transformers, even compared to ImageNet-21k supervised trained backbones. Among all the backbones, CLIP is only worse than ImageNet-21k trained SwinV2 and ConvNeXt, which shows the power of vision-language pretraining and again, suggests that we should consider more backbones other than plain ViTs when conducting self- or weakly-supervised learning.

▷ **What about generative backbones?** In contrast to models trained using supervised or self-supervised approaches with contrastive loss, backbones trained with a generative objective, such as MAE or Stable Diffusion, had comparatively inferior performance. We recommend caution when interpreting this result, as the evaluation of Stable Diffusion is currently limited to select tasks. Nonetheless, Stable Diffusion is a larger backbone than others considered in this benchmark and is trained on a very large dataset, yet it exhibits inferior performance.

▷ **Battle of the "small" backbones.** Keeping limited resources in mind, we also compare the "small" subset of backbones in BoB ($< 30$M parameters) – with ViT-Small, ConvNeXt-Tiny, Swin-Tiny and ResNet-50 architectures. Overall, we find Supervised ConvNeXt-T trained on IN-1k to be the best, followed by Supervised SwinV2-T trained on IN-1k and DINO ViT-S trained on IN-1k. Interestingly, supervised learning again dominates, and backbones pretrained on just IN-1k outperform ones trained on a considerably more diverse and larger dataset (MiDaS).

▷ **Performance vs. Speed?** Our analysis reveals a strong negative correlation ($\rho = -0.41$) between throughput (computed on NVIDIA RTX A5000) and average performance z-scores across all tasks when considering each backbone. This finding aligns with our previous observation that larger models tend to exhibit superior performance. Consequently, in order to achieve enhanced performance, one may need to sacrifice speed.

▷ **Monocular depth-estimation as a general purpose pretraining strategy.** In our experiments, MiDaS achieves performance competitive with that of top conventional supervised and SSL backbones at classification, object detection, and segmentation, even outside of the natural image domain, for example on satellite images. This observation suggests that depth-estimation may serve as a powerful and generalizable primary or auxiliary pretraining task for foundation models, supporting findings of Lao et al. [49].

▷ **Calibration and test likelihood are correlated with accuracy.** We measure expected calibration error (ECE) as well as test cross-entropy loss on the ImageNet test set. Whereas test likelihood is

strongly correlated with accuracy ($r = -0.8278$), ECE exhibits a weaker correlation ($r = -0.4876$). In both cases, we observe p-values under $0.05$. We also note that self-supervised pretraining typically leads to inferior calibration.

▷ **CNNs and SSL are more adversarially robust.** We additionally measure the adversarial robustness of each backbone on the ImageNet test set using an $\ell_\infty$-constrained PGD attack with multiple radii (see Appendix Table 19). For each architecture where we possess self-supervised learning versions, we see that supervised pretraining always yields inferior robustness. Moreover, ViTs are more vulnerable to adversarial examples than convolutional networks. Notably, ConvNeXt is more adversarially robust even when trained in a supervised fashion.

## 6 Where Are Things Going From Here?

At the core of every computer vision model is a backbone. In our battle of the backbones, we compared more than 1,500 training runs to surface insights for computer vision practitioners and researchers. To guide practitioners, we analyzed the performance of publicly available vision backbones across a broad range of tasks from segmentation and detection to classification and retrieval. We found supervised ConvNext, supervised SwinV2, and CLIP models performed well across this broad range of tasks. For computationally constrained settings, in our battle of the "small" backbones we found smaller counterparts to the same archiectures supervised ConvNext-T and SwinV2, followed by DINO with a small ViT performed quite well. BoB offers practitioners a guide to select sensible backbones from the dizzying array of choices.

For researchers looking ahead, we also observed several notable trends. First, we found performance across tasks is strongly correlated, suggesting a shift away from specialized vision backbones to universal backbones that work well across a range of tasks. Next, we found throughput and performance are inverse related, suggesting scaling remains a promising avenue to improve backbones. Finally, we found that while our practical recommendations include many supervised models, in apple-to-apples comparisons to standard supervised training, self-supervised learning holds promise. By releasing all our experimental results along with code to put new backbones to the test, we hope BoB serves as a useful guide to both practitioners today and researchers looking ahead at tomorrow.

**Limitations.** We note that insights obtained from BoB are contingent on the vocabulary of tasks, backbones, and settings considered in this work. We intend for takeaways from this study to provide practical considerations useful for computer vision researchers, recognizing that such insights need to continuously evolve as more backbones are introduced and more tasks and settings are taken into account. Lastly, we note that studies in BoB focus mostly primarily on aspects related to performance, and exploration along other axes of importance (biases in models, etc.) remain.

Our benchmark does not include backbones larger than ConvNext-Base, aside from Stable Diffusion, and some rankings may change at a large scale. For instance, while we find that modern convolutional architectures pretrained via supervised learning perform best on most tasks, we also find that transformers benefit more from scale, both in terms of pretraining data and architecture size. It is possible that transformer backbones will pull ahead of convolutional backbones at very large scales.

## 7 Computation Cost and Carbon Footprint

The experiments in this paper took a cumulative 127k GPU hours on `NVIDIA RTX A100` cards. Assuming the GPUs were running with an average carbon efficiency of 0.37 $kgCO_2eq/kWh$, the total emissions are estimated to be 11792.36 $kgCO_2eq$ [48].

## Acknowledgements

MG and AGW were supported in part by NSF CAREER IIS-2145492, NSF I-DISRE 193471, NIH R01DA048764-01A1, NSF IIS-1910266, BigHat Biosciences, Capital One, and an Amazon Research Award. HS and RC were supported in part by the ONR MURI grant N00014-20-1-2787. VP, PC, and JH were supported in part by ARL, NASA ULI, Google, and NSF #2144194. RN, MS, GS, and TG were supported by the ONR MURI program, the Office of Naval Research (N000142112557), the AFOSR MURI program, and the National Science Foundation (IIS-2212182 & 2229885).

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

# A   Additional Related Work

**Classification benchmarks.**   Image classification is the most common task in computer vision, and we have multiple benchmarks. For example, the `timm` library [100] benchmarks ImageNet classification performance across loads of backbones trained with different methods and on different datasets. In addition, we have dataset-specific benchmarks, such as "paperwithcode"[3]. The latter contains multiple datasets and methods, but it lacks systematic analysis among these results. Almost all the self-supervised learning method papers perform their own evaluation for image classification. To accelerate the research cycle in self-supervised learning, VISSL [26] provides a library for implementing SSL methods and evaluations. In this work, we evaluate various backbones trained in both self-supervised and supervised fashion, and on multiple datasets on different domains (natural images, satellite maps, and medical images). Moreover, we benchmark these backbones and datasets with exactly the same learning setting and conduct a thorough analysis of the collected results, which we believe is essential and helpful to practitioners.

**Object detection and segmentation benchmarks.**   Benchmarking backbones for object detection and segmentation has been an active area of research. Several works have focused on evaluating and comparing the performance of various backbone architectures for these tasks [51, 52, 29, 9]. Popular backbone networks such as supervised pretrained ResNet have been extensively utilized and compared, while modern feature extractors such as vision-language models and self-supervised learning models have not been extensively studied. These studies either focus on a limited subset of backbones or incorporate diverse detectors with varying backbones, thereby hindering an accurate comparison. To the best of our knowledge, we are the first to present a comprehensive study of the various backbones with various architectures and pretraining methods for object detection and instance segmentation.

**Out-of-distribution generalization benchmarks.**   Several prior works have benchmarked the out-of-distribution performance of visual models. For image classification, these have included variants of ImageNet [32–34, 93, 76, 63], synthetic-to-real adaptation benchmarks [66], and benchmarks with images belonging to varied domains including paintings, clipart, etc. [50, 67, 90, 87], sometimes even spanning multiple modalities and applications [43]. Similarly for object detection, several OOD generalization benchmarks have been fashioned from sim-to-real, cross-weather, and cross-city self-driving datasets [38, 15, 105, 80, 64]. Recently, [41] conducted a broad study of pretraining strategies for domain adaptation and generalization. In this work, we perform a similar study but on a larger scale and also include a diverse suite of backbones designed for varied tasks.

**Image retrieval benchmarks.**   To the best of our knowledge, our study represents the first comprehensive examination of multiple pretrained deep learning backbones for image retrieval task. While previous survey papers [11, 108, 20, 39] have explanations of various types of deep learning methods, such as off-the-shelf models trained in an unsupervised or supervised fashion, single pass, and multiple pass methods, none have **quantitatively** analyzed these backbones. Therefore, our work fills this crucial gap in the existing literature.

# B   An Extended Guide to BoB

## B.1   The Architectures

Below is a list of all architectures we compare. As different neural network architectures are believed to have distinct properties, from invariances to a reliance on different Fourier frequencies, evaluating a variety of architectures will allow us examine potential benefits of architectural differences. Many of the pretraining strategies we evaluate are accompanied by multiple checkpoints with different architectures or sizes, so we include multiple versions of each. We describe architectural modifications to these backbones for object detection and segmentation in Section 3.2.

- **ResNet [28]:** These are the staple convolutional neural networks we all know and love, complete with skip connections and batch normalization [36]. We include experiments on ResNet-50 and ResNet-101 backbones.

---

[3]https://paperswithcode.com/

- **ConvNeXt [58]:** ConvNeXt is a purely convolutional network with numerous modern architectural improvements including depthwise convolutions, inverted bottleneck blocks, large convolutional kernels, and a larger proportion of layers allocated to the third stage. This architecture improves performance of convolutional architectures at scale while maintaining their strong object detection and segmentation capabilities. We include experiments on ConvNeXt-Tiny and ConvNeXt-Base.

- **Vision Transformer [18]:** Vision transformers (ViTs) were derived from transformer language models [89] and inherit their multi-headed self-attention (MHSA) and position-wise feed-forward network (FFN) components. Unlike ResNets, ViTs do not encode translation equivariance, and they only encode locality by embedding images on a patch-by-patch basis. We include experiments on ViT-Small and ViT-Base.

- **Swin Transformer V2 [57]:** Swin Transformer [56] is a transformer architecture which incorporates hierarchical representations, translation invariance, and increased locality and efficiency into ViTs by only performing attention within spatial windows and merging these windows iteratively. SwinV2 is equipped with several modifications which improve scalability and transferability across input resolutions. We include experiments on SwinV2-Tiny and SwinV2-Base. For SwinV2-Base, unless otherwise stated, we use the model with a window size of 24.

- **Stable Diffusion encoder [77]:** Stable Diffusion is a text-to-image generative diffusion model which conducts diffusion in a latent space. We include experiments with a backbone formed by the learned encoder that converts images from pixel-space into the latent space where diffusion is performed followed by Stable Diffusion's U-Net, and we freeze the text encoder, using its frozen embedding. The encoder uses a convolutional architecture with added attention mechanisms. More details can be found in Rombach et al. [77].

## B.2 The Pretraining Algorithms

The primary source of diversity amongst the backbones we consider stems from their different pretraining algorithms. We choose prototypical examples of categories including supervised learning, self-supervised learning (SSL), and vision-language since such types of pretraining routines are widely believed to confer their own unique properties. For instance, SSL backbones are thought to extract more transferable features [86], while vision-language models are thought to resist domain shifts [78].

- **Classification:** We include image classifiers pretrained on ImageNet-1k and -21k [17]. ImageNet pretraining has long been the de facto choice for computer vision systems.

- **MiDaS [75]:** MiDaS is trained in a supervised fashion for monocular depth estimation. In this task, the model accepts a natural image and outputs a dense 2D array of depth values representing the distance of the scene from the image plane.

- **MoCo v3 [12]:** Contrastive learning is a popular approach to SSL in computer vision which encourages a model extract similar features corresponding to different augmentations of the same image, called *positive pairs* and dissimilar features corresponding to different images, called *negative pairs*. MoCo v3 is a high-performing variant which employs a momentum encoder and multi-crop training as well as prediction and projection heads.

- **VICReg [3]:** Instead of adopting contrastive learning and negative pairs, VICReg avoids feature vectors collapsing towards a constant during SSL by regularizing their variance and covariance. VICRegL is a version which also applies the VICReg criterion to local features to teach the model to extract localization information in its features for downstream dense prediction tasks [4].

- **DINO [8]:** Much like MoCo v3, DINO uses a momentum encoder and multi-crop SSL, but DINO swaps out contrastive learning for self-distillation, demonstrating strong performance on ViTs with a small patch size.

- **MAE [30]:** Masked Autoencoders (MAE) use a distinct style of SSL adapted from masked language modeling [40]. MAE models are trained to reconstruct masked out input patches, unlike the above 3 models which instead match the features of augmented images.

- **CLIP [73]:** CLIP also uses contrastive learning, but on image-caption pairs instead of augmented image views. Language supervision endows CLIP features with information relating to the semantic meaning of image components, compared to models trained solely on image data [22]. We only use CLIP's image feature extractor in our experiments.

- **Stable Diffusion [77]:** Text-to-image diffusion models are an entirely different type of vision-language backbone, trained for image generation. The Stable Diffusion encoder, which we benchmark, maps images to a highly compressed latent space where diffusion is performed.

- **Random initialization:** In experiments where we fine-tune backbones on downstream tasks, we also evaluate baselines trained on the downstream training sets from random initialization.

## B.3 The Pretraining Datasets

The backbones we benchmark are pretrained on datasets across a wide range of scales including image classification, image-text pairs, and images with depth annotations:

- **ImageNet-1k and -21k [17]:** ImageNet-21k contains over 14 million training images in 21,841 classes. ImageNet-1k is a subset of the aforementioned dataset containing almost 1.3 million training images in 1000 classes. These popular web-scraped image classification datasets are used for supervised pretraining with the labels, or self-supervised pretraining without them, among numerous backbones we benchmark. We denote pretraining on ImageNet-21k followed by fine-tuning on ImageNet-1k by "ImageNet-21k-1k".

- **LAION-2B [81]:** LAION-2B is a subset of the larger LAION-5B, which contains 5.85 billion web-scraped image-text pairs filtered by CLIP. LAION-2B specifically contains those 2.32 billion pairs with English language captions. Despite being by far the largest dataset amongst those we consider, LAION-2B is known to contain a large number of duplicates [97]. Stable Diffusion is further fine-tuned on LAION-Aesthetic, a subset of LAION-2B containing 120 million images filtered by a model trained to rate images by aesthetics.

- **CLIP [73]:** Since there is no OpenCLIP ResNet checkpoint available, we use the original CLIP checkpoint trained on OpenAI's diverse proprietary captioned-image dataset containing 400 million images scraped from publicly available internet sources.

- **MiDaS [75]:** MiDaS was trained on a combination of 12 image datasets with various types of depth annotations, and objectives: ReDWeb [101], DIML [42], Movies [75], MegaDepth [53], WSVD [92], TartanAir [96], HRWSI [102], ApolloScape [35], BlendedMVS [104], IRS [94], KITTI [62], NYU Depth V2 [82]. These models are therefore trained with multi-objective optimization. Collectively, the MiDaS training set contains more than 3.7 million images. An earlier version of MiDaS was trained on a smaller collection of 5 datasets, but we use the most recent version trained on the largest training set.

**Evaluation datasets and licenses.** In Table 4, Table 5, Table 6, and Table 7 we summarize the datasets we use for evaluating classification, object detection, segmentation, out-of-domain generalization and retrieval performance. We include the number of classes as well as the number of test samples for each dataset. To be noticed, we only use $10\%$ of the labeled dataset for EuroSAT and Chexpert to distinguish the performance among all the backbones. Object detection and instance segmentation experiments are conducted on COCO dateset [54]. COCO is released under the Creative Commons Attribution 4.0 License[4]. This license permits users to share and adapt the dataset for any purpose, as long as the original creators are appropriately credited. For OOD classification, we use the ImageNet-Adversarial [34], ImageNet-Sketch [93], ImageNet-Renditions [33], ImageNet-V2 [76], and VisDA [66] datasets, all of which are freely available for research use. For OOD detection, we use the Sim10k [38] and Cityscapes [15] datasets. Densely annotated images for Sim10k are available freely[5] and can only be used for non-commercial applications. The license agreement for the Cityscapes dataset dictates that the dataset is made freely available to academic and non-academic entities for non-commercial purposes such as academic research, teaching, scientific publications, or personal experimentation and that permission to use the data is granted under certain

---

[4]https://cocodataset.org/#termsofuse
[5]https://fcav.engin.umich.edu/projects/driving-in-the-matrix

Table 4: **Image Classification Datasets**

| Dataset | Description | Size | Classes |
|---|---|---|---|
| ImageNet-1k [17] | Natural images of versatile categories | 1.3M | 1,000 |
| CIFAR-100 [46] | Natural images of versatile categories | 50K | 100 |
| EuroSAT [31] | Satellite images (RGB) of land use and land cover | 13.5K | 10 |
| Flowers-102 [65] | Images of flowers categories | 1K | 102 |
| Aircraft [61] | Images of aircraft model variant, family, manufacturer | 3K | 100 |
| Chexpert [37] | Medical images | 191K | 5 |

Table 5: **Object Detection and Instance Segmentation Datasets**

| Dataset | Description | Size | Classes |
|---|---|---|---|
| COCO [54] | Large-scale object detection and segmentation dataset | 330K | 80 |

conditions.[6] All datasets used in benchmarking retrieval tasks (except for Objectnet) are restricted to non-commercial research and educational purposes. Objectnet is free to use for both research and commercial applications.[7]

## C   Experimental Details

### C.1   Implementation Details

**Classification.** For **fine-tuning**, we train the backbones for 100 epochs using AdamW [60] and weight decay $\{5e^{-2}, 1e^{-3}\}$. We use a cosine annealing learning rate scheduler with 5 warmup epochs. We run grid searches for learning rates with the default grid range being $\{1e^{-3}, 5e^{-4}, 1e^{-4}\}$ as we observe peaking performance on multiple models with these learning rates. We expand the search range for learning rate when training models from scratch (*i.e.*, fine-tuning from randomly initialized weights) to $\{1e^{-2}, 5e^{-3}, 1e^{-3}, 5e^{-4}, 1e^{-4}\}$. We keep the batch size of 1024 the same for all experiments and use gradient accumulation when Out-of-Memory occurs for large models (such as the Stable Diffusion encoder). For data augmentation, we follow He et al. [30], including random horizontal flips and crops, mixup, CutMix, and a RandAug [16] policy with hyperparameter (9, 0.5) corresponding to the number of transformations and magnitude. For regularization strategies, we apply the stochastic depths with a drop rate of 0.1, layer-wise learning rate decay of 0.75, and label smoothing with a rate of 0.1. For **linear probing**, again we follow He et al. [30], where we set weight decay to zero and disable Mix-Up, Cut-Mix, stochastic depth, or color jitter. We train the model with LARS optimizer with a batch size of 4096 for 90 epochs. For **fine-tuning on low-shot ImageNet**, we follow Cai et al. [6], where we use AdamW for all the transformer-based backbones and SGD for Convolution-only backbones. For transformer-based backbones, we use grid search among three peak learning rates of $\{1e^{-4}, 2.5e^{-4}, 5e^{-4}\}$ and two layer-decay rates of $0.65, 0.75$ for AdamW. We use grid search among three peak learning rates of $\{1e^{-1}, 5e^{-2}\}$ for SGD. We fix the

---

[6]https://www.cityscapes-dataset.com/license/
[7]https://objectnet.dev/download.html

Table 6: **OOD Generalization Datasets**

| Task | Train Dataset | Test Dataset | Test Size | Classes |
|---|---|---|---|---|
| Image Classification | ImageNet-1K [79] | ImageNet-A [34] | 7.5K | 0.2K |
| Image Classification | ImageNet-1K [79] | ImageNet-v2 [76] | 10K | 1K |
| Image Classification | ImageNet-1K [79] | ImageNet-R [33] | 30K | 0.2K |
| Image Classification | ImageNet-1K [79] | ImageNet-S [93] | 50K | 1K |
| Image Classification | VisDA-Synthetic [66] | VisDA-Real [66] | 55.4K | 12 |
| Object Detection | Sim10K [38] | Cityscapes [15] | 0.5K | 1 |

Table 7: **Image Retrieval Datasets**: Kindly note that the provided numbers are approximate references. For precise details, please consult the original research papers.

| Dataset | Description | Size | Classes |
|---|---|---|---|
| CUB-200 [91] | Fine-grained bird images | 12k | 200 |
| i-Naturalist [88] | Fine-grained species classification | 450k | >1k |
| Object Net [2] | Object recognition dataset with uncommon poses | 50k | 313 |
| INSTRE [95] | Fine-grained recognition data set of toys and house-hold objects | 28k | 200 |
| Google Landmarks V2 [98] | Landmarks across the world | >1 mil | >1k |
| Oxford [69] | Specific buildings from Oxford | 6400 | 12 |
| Paris [70] | Specific landmarks and buildings from Paris | 5000 | 11 |
| Copy Days [19] | Copy detection dataset of landscapes, buildings etc | 2,286 | n/a |

weight decay to be $0.05$ and use the same data augmentation as the regular **fine-tuning** for $10\%$ but without strong data augmentations, such as mix-up, cut-mix, and random erasing for the $1\%$ setup.

**Object Detection and Segmentation.** For the training of Cascade Mask R-CNN, we adopt the standard training settings previously employed in ConvNext [58], Swin [56], and ViTDet [52]. For all experiments, we utilize the AdamW optimizer [60] with weight decay of $0.05$, batch size of $16$, and we conduct a grid search for the learning rate, considering values of $\{8e^{-5}, 1e^{-4}, 2e^{-4}, 3e^{-4}\}$. Additionally, we employ a $3\times$ schedule, spanning a total of 36 epochs, and the learning rate decayed by a factor of 10 at epochs 27 and 33. In addition, we apply multi-scale training [7, 83], excluding ViT-based backbones and Stable Diffusion. For ViT-based backbones, inspired by the approach employed in VITDet along with the fact that ViT backbones perform poorly on detection without specially tailored training strategies, we use *large-scale jitter* (LSJ) with the image resolution of $1024 \times 1024$ and scale range of $[0.1, 2.0]$. In order to maintain fair comparisons across backbones, we minimize architecture-specific modifications. Thus, for our ViT backbones, we avoid implementing some ViTDet modifications such as "simple feature pyramid" instead of FPN. Instead, we employ an FPN that utilizes the final feature map, without employing stage division. It is worth noting, as highlighted in the ViTDet paper, the performance of "FPN, last-map" is inferior to the "simple feature pyramid". Additionally, we use the supervised training results from DeiT [85] for supervised ViT-S and ViT-B pretrained backbones. For Stable Diffusion, we use a single resolution of $512 \times 512$ (resizing the image such that the longer side is $512$) to overcome the significantly larger computational cost of this backbone compared to its competitors in our benchmark.

*A note on ViTDet and scale*: architectural modifications and very long training schedules can benefit ViTs in particular, as was found in Li et al. [52]. Similarly, Chen et al. [14] point out that ViTDet achieves stronger performance than their own work due to long and expensive training routines, behavior which stems from ViTs weak vision inductive bias. Additionally, in our analysis, we found that ViTs benefit more from scale, so ViTs might overtake other models at larger scales. Including large models in our benchmark, which includes many tasks and pretraining methods, would be prohibitively expensive, yet practitioners with sufficient computational resources for larger models, longer training routines, and architectural modifications may consider ViT backbones.

**Syn-to-Real Generalization.** For VisDA [66] Syn→Real, we report accuracy on the target domain (real) using models trained on the source domain (synthetic) for 12-way classification. For training, we use a learning rate of $1e^{-3}$ on 4 A40 GPUs with a batch size of 64 and report accuracy after 10 epochs. For object detection, we train the backbones outlined in Appendix B.1 with the Cascade-RCNN architecture. For training and fine-tuning backbones on Sim10k [38], we use a learning rate of $1e^{-4}$ on 4 A40 or RTX 6000 GPUs. To enable consistent evaluation across all backbones (CNNs and ViTs), we downsample Cityscapes [15] images to $1024 \times 512$ during evaluation. We train all models on the entirety of Sim10k and evaluate on the validation split of Cityscapes.

**Image Retrieval.** We have only evaluated pretrained models for this task. Dataset and metrics are discussed in the main body. Refer to table Table 7 for a brief summary of all the retrieval datasets.

# D   Results

**Image Classification.** We present the ImageNet Top-1 and Top-5 classification accuracy for backbones pretrained with different methods and on various datasets in Table 8. We adopt the ImageNet results for supervised learning with random initialization from the `timm` library [100]. We omit

ImageNet results for ImageNet-pretrained backbones since those coincide with the randomly initialized backbones on ImageNet. We also present top-1 classification accuracy for finetuning on various datasets in Table 9, and we include ImageNet calibration and uncertainty estimation results in Table 10.

Table 8: **Classification accuracy (%) for ImageNet-related tasks.** "lp" denotes linear probing, "1%" and "10%" denote the percentage of labeled training images used during the fine-tuning.

| Backbone | Method | Pretrain Data | ImageNet (lp) | | ImageNet | | ImageNet (10%) | | ImageNet (1%) | |
|---|---|---|---|---|---|---|---|---|---|---|
| | | | Top-1 | Top-5 | Top-1 | Top-5 | Top-1 | Top-5 | Top-1 | Top-5 |
| ResNet-50 | Supervised | ImageNet-1k | - | - | 80.38 | 94.60 | - | - | - | - |
| | VicReg | ImageNet-1k | 72.15 | 90.22 | 78.77 | 94.29 | 69.54 | 89.07 | 55.04 | 79.34 |
| | CLIP | LAION-2B | 65.98 | 87.84 | 80.55 | 95.26 | 69.26 | 89.91 | 43.78 | 70.67 |
| | DINO | ImageNet-1k | 74.17 | 91.56 | 79.08 | 94.60 | 68.18 | 89.35 | 51.38 | 77.82 |
| ViT-Small | Supervised | ImageNet-1k | - | - | 78.84 | 94.29 | - | - | - | - |
| | MoCoV3 | ImageNet-1k | 73.11 | 90.94 | 79.65 | 94.96 | 70.27 | 90.11 | 54.14 | 79.15 |
| | DINO | ImageNet-1k | 76.08 | 92.63 | 81.33 | 95.71 | 73.83 | 91.89 | 58.15 | 80.07 |
| ViT-Base | Supervised | ImageNet-1k | - | - | 79.15 | 94.09 | - | - | - | - |
| | MoCoV3 | ImageNet-1k | 75.96 | 92.69 | 82.85 | 96.31 | 74.80 | 92.54 | 62.88 | 85.31 |
| | MAE | ImageNet-1k | 67.67 | 87.49 | 83.41 | 96.50 | 72.87 | 91.54 | 56.02 | 81.07 |
| | DINO | ImageNet-1k | 77.31 | 93.43 | 83.40 | 96.42 | 75.92 | 93.30 | 63.92 | 84.52 |
| | CLIP | LAION-2B | 79.74 | 95.53 | 85.19 | 97.46 | 78.67 | 95.00 | 66.44 | 89.11 |
| SwinV2-Tiny | Supervised | ImageNet-1k | - | - | 81.82 | 95.99 | - | - | - | - |
| | MiDaS | MiDaS | 76.44 | 92.66 | 82.55 | 95.92 | 79.92 | 94.57 | 75.44 | 90.83 |
| SwinV2-Base | Supervised | ImageNet-21k-1k | - | - | 87.10 | 98.23 | - | - | - | - |
| | MiDaS | MiDaS | 81.09 | 95.47 | 86.48 | 98.00 | 84.14 | 96.91 | 79.26 | 93.66 |
| Stable Diffusion | Stable Diffusion | LAION-2B | - | - | 79.90 | 95.10 | 71.50 | 89.07 | 38.02 | 65.78 |

Table 9: **Top-1 classification accuracy (%) for fine-tuning pretrained (and randomly initialized) backbones with different methods on various datasets.** We omit the ImageNet performance for the backbones trained on ImageNet in a supervised fashion that setup is the same as backbones trained from random initialization.

| Backbone | Method | Pretrain Data | ImageNet | EuroSAT | Flower | CIFAR-100 | Chexpert | Aircraft |
|---|---|---|---|---|---|---|---|---|
| ResNet-50 | Rand Init | - | 80.38 | 89.61 | 41.96 | 72.33 | 86.67 | 20.73 |
| | Supervised | ImageNet-1k | - | 98.26 | 86.52 | 84.71 | 86.82 | 55.31 |
| | VicReg | ImageNet-1k | 78.77 | 95.11 | 92.68 | 87.56 | 86.55 | 67.51 |
| | CLIP | LAION-2B | 80.55 | 98.72 | 90.62 | 84.35 | 88.92 | 73.50 |
| | DINO | ImageNet-1k | 79.08 | 98.74 | 94.04 | 86.49 | 87.49 | 74.13 |
| ResNet-101 | Rand Init | - | 81.93 | 62.07 | 44.51 | 67.94 | 87.41 | 13.08 |
| | Supervised | ImageNet-1k | - | 97.19 | 83.59 | 83.07 | 86.51 | 40.47 |
| ViT-Small | Rand Init | - | 78.84 | 42.61 | 38.73 | 56.08 | 77.36 | 5.52 |
| | Supervised | ImageNet-1k | - | 95.54 | 96.17 | 89.48 | 87.60 | 61.69 |
| | Supervised | ImageNet-21k | 81.39 | 95.30 | 98.73 | 92.51 | 87.39 | 59.89 |
| | MoCoV3 | ImageNet-1k | 79.65 | 96.02 | 83.59 | 89.38 | 88.10 | 54.68 |
| | DINO | ImageNet-1k | 81.33 | 89.09 | 73.73 | 90.00 | 87.75 | 48.20 |
| ViT-Base | Rand Init | - | 79.15 | 48.33 | 40.20 | 54.30 | 79.61 | 6.99 |
| | Supervised | ImageNet-1k | - | 96.22 | 94.04 | 90.62 | 87.27 | 62.02 |
| | Supervised | ImageNet-21k | 84.53 | 94.93 | 99.41 | 93.89 | 87.72 | 70.2 |
| | MoCoV3 | ImageNet-1k | 82.85 | 96.74 | 94.53 | 90.89 | 86.82 | 71.55 |
| | MAE | ImageNet-1k | 83.41 | 95.54 | 94.63 | 89.96 | 88.01 | 72.27 |
| | DINO | ImageNet-1k | 83.40 | 95.59 | 97.17 | 91.22 | 87.05 | 71.82 |
| | CLIP | LAION-2B | 85.19 | 94.37 | 96.78 | 91.29 | 87.74 | 76.38 |
| SwinV2-Tiny | Rand Init | - | 81.82 | 89.33 | 29.90 | 66.49 | 87.76 | 5.31 |
| | Supervised | ImageNet-1k | - | 98.91 | 96.58 | 89.50 | 88.39 | 68.71 |
| | MiDaS | MiDaS | 82.55 | 96.33 | 96.48 | 90.53 | 87.69 | 69.54 |
| SwinV2-Base | Supervised | ImageNet-21k-1k | 87.10 | 95.94 | 99.61 | 93.09 | 87.73 | 79.46 |
| | MiDaS | MiDaS | 86.48 | 96.53 | 99.32 | 93.08 | 88.08 | 79.62 |
| ConvNeXt-Tiny | Rand Init | - | 82.10 | 73.80 | 18.24 | 75.31 | 83.25 | 4.35 |
| | Supervised | ImageNet-1k | - | 95.70 | 96.00 | 89.89 | 87.56 | 69.99 |
| ConvNeXt-Base | Rand Init | - | 83.88 | 30.39 | 19.41 | 73.04 | 85.26 | 4.02 |
| | Supervised | ImageNet-1k | - | 93.06 | 94.92 | 88.98 | 88.98 | 64.72 |
| | Supervised | ImageNet-21k | 85.87 | 96.19 | 99.61 | 92.84 | 87.80 | 69.39 |
| Stable Diffusion | Stable Diffusion | LAION-2B | 79.90 | 92.22 | 91.89 | 90.50 | 87.49 | 72.45 |

**Object Detection and Segmentation.** In our experiment, we utilize the *train2017* and *val2017* splits of the COCO dataset for training and evaluation, respectively. We report results on bounding box object detection ($AP^{box}$) and instance segmentation ($AP^{mask}$). In Table 11, Table 12 and Table 13, we present the comprehensive results of our experiment. Table 11 reflects the results obtained with various backbone architectures when the detector is fine-tuned while the backbone remains frozen. Table 12 contains both training from scratch with randomly initialized backbones as well as end-to-end fine-tuning using pretrained backbones. In Table 13, we additionally present results on the harder

Table 10: **Calibration and uncertainty estimation on ImageNet.** We measure Top-1 accuracy (%), cross-entropy test loss (**CE**), and expected calibration error (**ECE**).

| Backbone | Method | Pretrain Data | Accuracy | CE | ECE |
|---|---|---|---|---|---|
| ResNet-50 | Supervised | ImageNet-1k | 80.38 | 0.94 | 0.09 |
| | VicReg | ImageNet-1k | 78.77 | 1.11 | 0.21 |
| | CLIP | LAION-2B | 80.55 | 1.02 | 0.19 |
| | DINO | ImageNet-1k | 79.08 | 1.11 | 0.22 |
| ResNet-101 | Supervised | ImageNet-1k | 81.93 | 0.92 | 0.16 |
| ViT-Small | Supervised | ImageNet-1k | 78.84 | 0.84 | 0.03 |
| | Supervised | ImageNet-21k | 81.39 | 0.68 | 0.01 |
| | MoCoV3 | ImageNet-1k | 79.65 | 0.90 | 0.11 |
| | DINO | ImageNet-1k | 81.33 | 0.83 | 0.10 |
| ViT-Base | Supervised | ImageNet-1k | 79.15 | 0.86 | 0.05 |
| | Supervised | ImageNet-21k | 84.53 | 0.56 | 0.01 |
| | MoCoV3 | ImageNet-1k | 82.85 | 0.77 | 0.08 |
| | MAE | ImageNet-1k | 83.41 | 0.75 | 0.09 |
| | DINO | ImageNet-1k | 83.40 | 0.76 | 0.07 |
| | CLIP | LAION-2B | 85.19 | 0.66 | 0.08 |
| SwinV2-Tiny | Supervised | ImageNet-1k | 81.82 | 0.83 | 0.09 |
| | MiDaS | MiDaS | 82.55 | 0.83 | 0.07 |
| ConvNeXt-Tiny | Supervised | ImageNet-1k | 82.10 | 0.79 | 0.06 |
| ConvNeXt-Base | Supervised | ImageNet-1k | 83.88 | 0.69 | 0.04 |
| | Supervised | ImageNet-21k | 85.87 | 0.56 | 0.03 |

LVIS dataset [27] using the best transformer-based (SwinV2) and convolutional-based (ConvNeXt) architectures from our experiments at several sizes. We again see a benefit of scale here as well as the slightly superior performance of modern convolutional architectures. These tables provide a comprehensive overview of the performance achieved across various backbones and scenarios, enabling a thorough analysis and comparison of the different backbones utilized in our study.

Table 11: **Object detection and instance segmentation using Cascade Mask-RCNN on COCO with frozen backbones.**

| Backbone | Method | Pretrain Data | Params | Input Size | $AP^{box}$ | $AP^{box}_{50}$ | $AP^{box}_{75}$ | $AP^{mask}$ | $AP^{mask}_{50}$ | $AP^{mask}_{75}$ |
|---|---|---|---|---|---|---|---|---|---|---|
| ResNet-50 | Supervised | ImageNet-1k | 82M | $1333 \times 800$ | 42.5 | 61.0 | 46.2 | 37.1 | 58.1 | 39.5 |
| | VicReg | ImageNet-1k | 82M | $1333 \times 800$ | 44.1 | 62.3 | 48.1 | 38.8 | 59.7 | 42.0 |
| | DINO | ImageNet-1k | 82M | $1333 \times 800$ | 44.6 | 62.9 | 48.8 | 39.0 | 60.3 | 42.0 |
| ResNet-101 | Supervised | ImageNet-1k | 101M | $1333 \times 800$ | 43.4 | 62.1 | 47.1 | 38.0 | 59.3 | 40.7 |
| ViT-Small | Supervised | ImageNet-1k | 84M | $1024 \times 1024$ | 27.3 | 44.3 | 29.1 | 23.6 | 40.8 | 23.9 |
| | MoCoV3 | ImageNet-1k | 84M | $1024 \times 1024$ | 27.8 | 44.6 | 29.7 | 24.2 | 41.5 | 24.9 |
| | DINO | ImageNet-1k | 84M | $1024 \times 1024$ | 33.6 | 52.5 | 35.9 | 29.1 | 49.0 | 30.0 |
| ViT-Base | Supervised | ImageNet-1k | 155M | $1024 \times 1024$ | 34.1 | 54.0 | 36.1 | 29.6 | 50.6 | 30.0 |
| | MoCoV3 | ImageNet-1k | 155M | $1024 \times 1024$ | 32.1 | 50.4 | 34.6 | 28.2 | 47.1 | 29.3 |
| | MAE | ImageNet-1k | 155M | $1024 \times 1024$ | 35.6 | 54.0 | 38.7 | 31.8 | 51.1 | 33.7 |
| | DINO | ImageNet-1k | 155M | $1024 \times 1024$ | 36.2 | 55.6 | 39.1 | 31.7 | 52.2 | 32.9 |
| | CLIP | LAION-2B | 155M | $1024 \times 1024$ | 21.9 | 36.7 | 22.5 | 18.8 | 33.6 | 18.8 |
| SwinV2-Tiny | Supervised | ImageNet-1k | 86M | $1333 \times 800$ | 36.9 | 55.4 | 39.9 | 32.5 | 52.6 | 34.4 |
| | MiDaS | MiDaS | 86M | $1333 \times 800$ | 34.3 | 51.5 | 37.2 | 30.2 | 48.9 | 32.0 |
| SwinV2-Base-w8 | Supervised | ImageNet-1k | 145M | $1333 \times 800$ | 38.6 | 57.4 | 41.7 | 33.5 | 54.4 | 35.3 |
| SwinV2-Base-w24 | Supervised | ImageNet-21k-1k | 145M | $1333 \times 800$ | 44.6 | 64.7 | 48.6 | 38.8 | 61.6 | 41.5 |
| | MiDaS | MiDaS | 145M | $1333 \times 800$ | 42.2 | 61.5 | 45.9 | 37.1 | 58.7 | 39.3 |
| ConvNeXt-Tiny | Supervised | ImageNet-1k | 86M | $1333 \times 800$ | 44.4 | 63.1 | 48.7 | 38.7 | 60.7 | 41.6 |
| ConvNeXt-Base | Supervised | ImageNet-1k | 146M | $1333 \times 800$ | 44.8 | 64.1 | 48.6 | 39.2 | 61.4 | 42.0 |
| | Supervised | ImageNet-21k | 146M | $1333 \times 800$ | 46.2 | 66.0 | 50.2 | 40.1 | 62.9 | 43.0 |
| Stable Diffusion | Stable Diffusion | LAION-2B | 442M | $1333 \times 800$ | 38.2 | 55.4 | 41.6 | 34.0 | 52.7 | 36.4 |

Table 12: **Object detection and instance segmentation results using Cascade Mask-RCNN on COCO.**

| Backbone | Method | Pretrain Data | Params | Input Size | $\text{AP}^{box}$ | $\text{AP}^{box}_{50}$ | $\text{AP}^{box}_{75}$ | $\text{AP}^{mask}$ | $\text{AP}^{mask}_{50}$ | $\text{AP}^{mask}_{75}$ |
|---|---|---|---|---|---|---|---|---|---|---|
| ResNet-50 | Random Init | - | 82M | $1333 \times 800$ | 41.3 | 57.4 | 45.1 | 36.0 | 55.1 | 38.9 |
| | Supervised | ImageNet-1k | 82M | $1333 \times 800$ | 46.6 | 64.6 | 50.6 | 40.2 | 61.9 | 43.5 |
| | VicReg | ImageNet-1k | 82M | $1333 \times 800$ | 38.2 | 55.2 | 41.5 | 33.5 | 52.9 | 35.9 |
| | DINO | ImageNet-1k | 82M | $1333 \times 800$ | 45.4 | 63.5 | 49.2 | 39.4 | 61.1 | 42.4 |
| ResNet-101 | Random Init | - | 101M | $1333 \times 800$ | 45.7 | 63.0 | 50.0 | 39.5 | 60.5 | 43.1 |
| | Supervised | ImageNet-1k | 101M | $1333 \times 800$ | 47.7 | 65.6 | 52.0 | 41.3 | 63.2 | 44.6 |
| ViT-Small | Random Init | - | 84M | $1024 \times 1024$ | 43.2 | 62.2 | 47.2 | 38.0 | 59.3 | 40.7 |
| | Supervised | ImageNet-1k | 84M | $1024 \times 1024$ | 48.2 | 68.1 | 51.8 | 41.7 | 65.0 | 44.7 |
| | MoCoV3 | ImageNet-1k | 84M | $1024 \times 1024$ | 47.6 | 67.5 | 51.7 | 41.6 | 64.3 | 44.4 |
| | DINO | ImageNet-1k | 84M | $1024 \times 1024$ | 48.2 | 67.6 | 52.4 | 42.3 | 64.9 | 44.9 |
| ViT-Base | Random Init | - | 155M | $1024 \times 1024$ | 46.0 | 65.0 | 50.7 | 40.1 | 62.3 | 43.1 |
| | Supervised | ImageNet-1k | 155M | $1024 \times 1024$ | 49.4 | 69.3 | 53.5 | 42.9 | 66.1 | 46.2 |
| | MoCoV3 | ImageNet-1k | 155M | $1024 \times 1024$ | 49.7 | 69.4 | 53.9 | 43.2 | 66.6 | 46.5 |
| | MAE | ImageNet-1k | 155M | $1024 \times 1024$ | 51.3 | 70.3 | 55.9 | 44.5 | 67.7 | 48.1 |
| | DINO | ImageNet-1k | 155M | $1024 \times 1024$ | 49.5 | 69.0 | 53.7 | 42.8 | 66.1 | 46.0 |
| | CLIP | LAION-2B | 155M | $1024 \times 1024$ | 50.0 | 69.3 | 54.4 | 43.3 | 66.3 | 46.6 |
| SwinV2-Tiny | Random Init | - | 86M | $1333 \times 800$ | 47.0 | 65.3 | 51.2 | 40.8 | 62.7 | 44.1 |
| | Supervised | ImageNet-1k | 86M | $1333 \times 800$ | 50.2 | 69.1 | 54.6 | 43.4 | 66.3 | 46.9 |
| | MiDaS | MiDaS | 86M | $1333 \times 800$ | 50.2 | 69.3 | 54.5 | 43.5 | 66.4 | 47.0 |
| SwinV2-Base-w8 | Random Init | - | 145M | $1333 \times 800$ | 48.1 | 66.6 | 52.3 | 41.5 | 64.1 | 44.7 |
| | Supervised | ImageNet-1k | 145M | $1333 \times 800$ | 52.4 | 71.0 | 57.1 | 45.2 | 68.6 | 49.1 |
| SwinV2-Base-w24 | Supervised | ImageNet-21k-1k | 145M | $1333 \times 800$ | 52.9 | 71.4 | 57.5 | 45.7 | 69.0 | 49.6 |
| | MiDaS | MiDaS | 145M | $1333 \times 800$ | 52.7 | 71.4 | 57.2 | 45.7 | 69.0 | 49.7 |
| ConvNeXt-Tiny | Random Init | - | 86M | $1333 \times 800$ | 47.5 | 65.5 | 51.7 | 41.2 | 63.0 | 44.3 |
| | Supervised | ImageNet-1k | 86M | $1333 \times 800$ | 49.9 | 68.4 | 54.3 | 43.2 | 66.0 | 46.8 |
| ConvNeXt-Base | Random Init | - | 146M | $1333 \times 800$ | 48.3 | 66.4 | 52.7 | 41.9 | 63.8 | 45.3 |
| | Supervised | ImageNet-1k | 146M | $1333 \times 800$ | 51.7 | 70.2 | 56.0 | 44.6 | 67.7 | 48.3 |
| | Supervised | ImageNet-21k | 146M | $1333 \times 800$ | 52.9 | 71.7 | 57.3 | 45.8 | 69.2 | 49.9 |
| Stable Diffusion | Random Init | - | 442M | $1333 \times 800$ | 37.1 | 51.4 | 40.1 | 31.9 | 43.7 | 31.2 |
| | Stable Diffusion | LAION-2B | 442M | $1333 \times 800$ | 43.4 | 59.1 | 46.3 | 38.1 | 56.9 | 40.2 |

Table 13: **Object detection and instance segmentation using Cascade Mask-RCNN on LVIS v1.**

| Backbone | Method | Pretrain Data | Params | Input Size | $\text{AP}^{box}$ | $\text{AP}^{box}_{50}$ | $\text{AP}^{box}_{75}$ | $\text{AP}^{mask}$ | $\text{AP}^{mask}_{50}$ | $\text{AP}^{mask}_{75}$ |
|---|---|---|---|---|---|---|---|---|---|---|
| SwinV2-Tiny | Supervised | ImageNet-1k | 86M | $1333 \times 800$ | 33.0 | 46.3 | 35.3 | 29.9 | 44.5 | 31.9 |
| | MiDaS | MiDaS | 86M | $1333 \times 800$ | 32.6 | 45.7 | 34.9 | 29.6 | 43.9 | 32.0 |
| SwinV2-Base-w8 | Supervised | ImageNet-1k | 145M | $1333 \times 800$ | 35.7 | 48.7 | 38.0 | 32.0 | 47.0 | 34.4 |
| ConvNeXt-Tiny | Supervised | ImageNet-1k | 86M | $1333 \times 800$ | 33.2 | 46.1 | 35.4 | 29.9 | 44.3 | 32.2 |
| ConvNeXt-Base | Supervised | ImageNet-1k | 146M | $1333 \times 800$ | 35.8 | 48.8 | 38.0 | 32.0 | 46.9 | 34.5 |

**OOD Generalization.** We include results for OOD generalization for image classification in Table 14 and for object detection in Table 15.

Table 14: **Top-1 OOD classification accuracy** (%).

| Backbone | Method | Pretrain Data | Test Dataset | | | | Train Dataset | Test Dataset |
|---|---|---|---|---|---|---|---|---|
| | | | ImageNet-A | ImageNet-S | ImageNet-R | ImageNet-V2 | | VisDA (real) |
| ResNet-50 | Rand Init | - | - | - | - | - | VisDA (syn) | 22.07 |
| | Supervised | ImageNet-1k | 10.03 | 29.63 | 40.25 | 68.75 | VisDA (syn) | 47.77 |
| | VicReg | ImageNet-1k | 10.29 | 28.63 | 39.98 | 67.56 | VisDA (syn) | 44.19 |
| | CLIP | LAION-2B | 15.53 | 30.10 | 42.44 | 69 | VisDA (syn) | 21.98 |
| | DINO | ImageNet-1k | 10.13 | 28.38 | 39.74 | 67.55 | VisDA (syn) | - |
| ResNet-101 | Rand Init | - | - | - | - | - | VisDA (syn) | 21.81 |
| | Supervised | ImageNet-1k | 19.04 | 34.73 | 45.10 | 70.88 | VisDA (syn) | 56.56 |
| ViT-Base | Rand Init | - | - | - | - | - | VisDA (syn) | 22.16 |
| | Supervised | ImageNet-1k | 14.85 | 27.99 | 38.02 | 66.45 | VisDA (syn) | 63.57 |
| | Supervised | ImageNet-21k-1k | 43.24 | 43.21 | 56.79 | 74.01 | VisDA (syn) | 65.62 |
| | MoCoV3 | ImageNet-1k | 32.57 | 37.33 | 34.85 | 72.71 | VisDA (syn) | 57.21 |
| | MAE | ImageNet-1k | 36.2 | 34.75 | 48.77 | 73.47 | VisDA (syn) | 48.90 |
| | DINO | ImageNet-1k | 35.32 | 36.52 | 49.15 | 72.71 | VisDA (syn) | 59.96 |
| | CLIP | LAION-2B | 46.59 | 63.33 | 50.17 | 75.39 | VisDA (syn) | 46.05 |
| SwinV2-Tiny | MiDaS | MiDaS | 30.87 | 29.41 | 41.26 | 71.27 | VisDA (syn) | 61.62 |
| SwinV2-Base | Supervised | ImageNet-21k-1k | 45.97 | 37.00 | 49.69 | 74.42 | VisDA (syn) | 68.84 |
| | MiDaS | MiDaS | 63.05 | 48.77 | 63.12 | 77.06 | VisDA (syn) | 68.66 |
| ConvNeXt-Base | Supervised | ImageNet-1k | 41.92 | 37.29 | 51.07 | 74.09 | VisDA (syn) | 71.12 |
| | Supervised | ImageNet-21k | 60.8 | 48.92 | 62.53 | 76.96 | VisDA (syn) | 74.96 |

Table 15: **OOD object detection on Sim10k→Cityscapes.** Out-of-distribution generalization across backbones for object detection (Cascade-RCNN) models trained on Sim10k and evaluated on Cityscapes to detect instances of "car". (Frozen) column corresponds to settings where the backbone has been frozen.

| Backbone | Method | Pretrain Data | mAP@50 | mAP@50 (Frozen) |
|---|---|---|---|---|
| ResNet-50 | Random Init | – | 30.6 | – |
| | Supervised | ImageNet-1k | 46.5 | 52.4 |
| | VicReg | ImageNet-1k | 44.4 | 49.8 |
| | DINO | ImageNet-1k | 50.0 | 52.2 |
| ResNet-101 | Random Init | – | 25.2 | – |
| | Supervised | ImageNet-1k | 46.9 | 52.6 |
| ViT-Small | Random Init | – | 8.7 | – |
| | Supervised | ImageNet-1k | 33.7 | 15.9 |
| | MoCoV3 | ImageNet-1k | 32.8 | 19.4 |
| | DINO | ImageNet-1k | 43.0 | 31.3 |
| ViT-Base | Random Init | – | 10.3 | – |
| | Supervised | ImageNet-1k | 38.2 | 24.6 |
| | MoCoV3 | ImageNet-1k | 36.7 | 27.9 |
| | MAE | ImageNet-1k | 44.2 | 32.3 |
| | DINO | ImageNet-1k | 41.7 | 35.1 |
| | CLIP | LAION-2B | 38.1 | 9.7 |
| SwinV2-Tiny | Random Init | – | 40.9 | – |
| | Supervised | ImageNet-1k | 48.0 | 48.2 |
| | MiDaS | MiDaS | 49.5 | 45.5 |
| SwinV2-Base-w8 | Random Init | – | 38.0 | – |
| | Supervised | ImageNet-1k | 51.1 | 49.3 |
| SwinV2-Base-w24 | Supervised | ImageNet-21k-1k | 51.0 | 50.2 |
| | MiDaS | MiDas | 52.5 | 50.3 |
| ConvNeXt-Tiny | Random Init | – | 36.2 | – |
| | Supervised | ImageNet-1k | 54.5 | 50.3 |
| ConvNeXt-Base | Random Init | – | 25.5 | – |
| | Supervised | ImageNet-1k | 55.5 | 52.1 |
| | Supervised | ImageNet-21k | 52.4 | 53.9 |

**Retrieval.** We present the mAP, MRR and Recall@5 values for various backbones across different datasets in Table 16, Table 17, and Table 18, respectively.

**Analysis.** In Table 20, we compare the performance of backbones pretrained with SSL (including CLIP) and supervised learning on ImageNet-1k and ImageNet-21k. We pick the top-3 backbones in each category and calculate the mean z-scores for all the tasks.

**Adversarial robustness.** In Table 19, we show adversarial robustness on the ImageNet test set against an untargeted PGD adversarial attack $\epsilon$ with $\ell_\infty$ constraints of $\frac{1}{255}$ and $\frac{2}{255}$. For attack hyperparameters, we use 20 steps and step size $\frac{\epsilon}{5}$.

# E   Assets and Licenses

The assets used in this work can be categorized as – Code Repositories, Backbones and Dependencies (licenses for datasets are included in Appendix B.3).

Table 16: **MAP scores for image retrieval experiments.**

| Backbone | Method | Pretrain Data | CUB | iNat | Obj | INSTRE | GLM | rOxf | rPar | CopyD |
|---|---|---|---|---|---|---|---|---|---|---|
| ConvNext-Base | Supervised | ImageNet-1k | 20.55 | 5.53 | 11.42 | 52.13 | 12.71 | 26.73 | 55.84 | 79.32 |
| | Supervised | ImageNet-21k-1k | 62.51 | 19.43 | 19.57 | 69.4 | 21.34 | 42.42 | 73.33 | 86.18 |
| ConvNext-Small | Supervised | ImageNet-1k | 23.96 | 5.66 | 10.35 | 49.33 | 12.6 | 28.1 | 56.03 | 77.86 |
| | Supervised | ImageNet-21k-1k | 61.09 | 17.06 | 17.12 | 62.79 | 19.53 | 42.19 | 71.03 | 83.56 |
| ConvNext-XLarge | Supervised | ImageNet-21k-1k | 65.31 | 21.98 | 21.37 | 69.28 | 21.67 | 45.33 | 74.76 | 86.81 |
| ResNet-101 | CLIP | OpenAI | 21.75 | 4.50 | 5.4 | 56.71 | 15.35 | 23.93 | 52.1 | 79.07 |
| | Supervised | ImageNet-1k | 22.86 | 5.52 | 8.9 | 47.37 | 13.83 | 31.27 | 58.51 | 78.52 |
| ResNet-50 | CLIP | OpenAI | 15.37 | 3.45 | 3.81 | 48.10 | 14.20 | 24.09 | 52.14 | 79.61 |
| | Supervised | ImageNet-1k | 18.75 | 4.28 | 5.31 | 41.59 | 11.98 | 25.59 | 52.37 | 79.78 |
| ResNet-50 | VicReg | ImageNet-1k | 8.84 | 3.15 | 1.82 | 42.45 | 13.76 | 31.71 | 58.24 | 82.48 |
| ResNet-50x64 | CLIP | OpenAI | 42.30 | 10.02 | 15.11 | 85.38 | 26.91 | 41.93 | 70.27 | 91.36 |
| SwinV2-Base | MiDaS | MiDaS | 10.44 | 3.84 | 1.93 | 24.91 | 11.52 | 29.76 | 56.34 | 84.03 |
| | Supervised | ImageNet-21k-1k | 57.57 | 17.95 | 17.81 | 66.87 | 20.33 | 44.35 | 71.28 | 87.57 |
| SwinV2-Base-w16 | Supervised | ImageNet-21k-1k | 58.77 | 19.93 | 18.6 | 69.48 | 20.97 | 44.91 | 71.7 | 88.2 |
| SwinV2-Large-w16 | Supervised | ImageNet-21k-1k | 57.55 | 17.74 | 16.36 | 71 | 21.64 | 44.26 | 75.58 | 88.59 |
| SwinV2-Tiny | MiDaS | MiDaS | 10.44 | 3.84 | 1.93 | 24.91 | 11.52 | 29.76 | 56.34 | 84.03 |
| SwinV2-Tiny | Supervised | ImageNet-1k | 22.91 | 5.71 | 6.94 | 54.31 | 13.76 | 27.35 | 56.29 | 82.93 |
| | Supervised | ImageNet-1k | 22.91 | 5.71 | 6.94 | 54.31 | 13.76 | 27.35 | 56.29 | 82.93 |
| ViT-Base | CLIP | LAION-2B | 40.27 | 8.17 | 12.98 | 81.13 | 23.83 | 44.84 | 73.62 | 88.79 |
| | MAE | ImageNet-1k | 1.26 | 0.22 | 0.43 | 4.86 | 1.43 | 5.19 | 11.12 | 48.12 |
| ViT-Base | MoCoV3 | ImageNet-1k | 12.97 | 4.16 | 2.29 | 40.53 | 12.11 | 30.25 | 51.6 | 84.75 |
| | Supervised | ImageNet-1k | 17.09 | 3.81 | 4.59 | 39.75 | 9.64 | 21.97 | 50.03 | 77.7 |
| ViT-Base | Supervised | ImageNet-21k-1k | 52.18 | 11.99 | 7.56 | 46.58 | 14.3 | 30.96 | 59.82 | 83.65 |
| | DINO | ImageNet-1k | 22.21 | 7.43 | 4.57 | 53.01 | 15.35 | 37.03 | 62.22 | 86.18 |
| ViT-Large | CLIP | LAION-2B | 47.77 | 9.62 | 12.87 | 80.29 | 25.45 | 39.19 | 70 | 90.32 |
| | MAE | ImageNet-1k | 2.53 | 0.84 | 0.53 | 12.24 | 3.78 | 13.26 | 24.92 | 70.34 |
| ViT-Large | Supervised | ImageNet-21k-1k | 62.44 | 18.19 | 16 | 55.18 | 18.49 | 37.68 | 67.07 | 87.04 |
| Vit-Small | MoCoV3 | ImageNet-1k | 12.99 | 3.7 | 1.86 | 35.28 | 11.39 | 24.83 | 50.24 | 82.67 |
| | Supervised | ImageNet-1k | 19.44 | 4.05 | 4.61 | 40.45 | 10.51 | 21.69 | 49.08 | 79.54 |
| Vit-Small | Supervised | ImageNet-21k-1k | 49.6 | 10.22 | 6.34 | 48.17 | 14.37 | 31.24 | 61.5 | 81.92 |
| | DINO | ImageNet-1k | 31.38 | 7.35 | 3.99 | 52.64 | 14.79 | 37.98 | 61.01 | 85.3 |

**Code Repositories.** We provide supporting code for all our experiments here. For image classification experiments, we build on top of the `timm` library [100][8], the original MAE repo[9] and the medical dataset pretrain repo [103][10]. `timm` is distributed under Apache 2.0 License and MAE under the Attribution-NonCommercial 4.0 International License. For object detection, instance segmentation, and OOD detection experiments, we build on top of the MMDetection framework [9][11]. MMDetection is distributed under Apache License 2.0.

**Backbones.** We use publicly available pretrained backbones. The full list is provided in Appendix B.

**Dependencies.** Key dependencies for all our experiments include `pytorch`, `timm`, HuggingFace utilities and MMCV. Please see our repo README for a comprehensive list of all dependencies to reproduce the experiments.

# F  Observations about Hyperparameters

For hyperparameter tuning, we find that the learning rate strategy is highly method- and dataset-dependent. For example, on ImageNet classification, the best learning rate we tried for CLIP was 1e-4 while the best learning rate for MAE was 1e-3, which is similar to the best learning fate for training from scratch. We speculate that this learning rate sensitivity occurs because different pretraining algorithms lead to parameter vectors of very different magnitudes. For image classification, a shorter

---

[8]https://github.com/huggingface/pytorch-image-models
[9]https://github.com/facebookresearch/mae
[10]https://github.com/lambert-x/Medical_MAE
[11]https://github.com/open-mmlab/mmdetection

Table 17: **MRR scores for image retrieval experiments.**

| Backbone | Method | Pretrain Data | CUB | iNat | Obj | INSTRE | GLM | rOxf | rPar | CopyD |
|---|---|---|---|---|---|---|---|---|---|---|
| ConvNext-Base | Supervised | ImageNet-1k | 0.63 | 0.15 | 0.38 | 0.88 | 0.37 | 0.63 | 0.97 | 0.87 |
| | Supervised | ImageNet-21k-1k | 0.9 | 0.41 | 0.52 | 0.94 | 0.53 | 0.84 | 0.99 | 0.92 |
| ConvNext-Small | Supervised | ImageNet-1k | 0.66 | 0.16 | 0.38 | 0.87 | 0.36 | 0.68 | 0.97 | 0.86 |
| | Supervised | ImageNet-21k-1k | 0.9 | 0.38 | 0.49 | 0.92 | 0.49 | 0.85 | 0.98 | 0.9 |
| ConvNext-XLarge | Supervised | ImageNet-21k-1k | 0.9 | 0.43 | 0.53 | 0.93 | 0.52 | 0.87 | 0.99 | 0.92 |
| ResNet-101 | CLIP | OpenAI | 0.67 | 0.15 | 0.31 | 0.93 | 0.44 | 0.65 | 0.97 | 0.86 |
| | Supervised | ImageNet-1k | 0.66 | 0.16 | 0.35 | 0.86 | 0.4 | 0.67 | 0.97 | 0.86 |
| ResNet-50 | CLIP | OpenAI | 0.59 | 0.12 | 0.24 | 0.88 | 0.41 | 0.67 | 0.97 | 0.86 |
| | Supervised | ImageNet-1k | 0.6 | 0.13 | 0.28 | 0.84 | 0.37 | 0.65 | 0.97 | 0.87 |
| | VicReg | ImageNet-1k | 0.45 | 0.11 | 0.18 | 0.86 | 0.42 | 0.8 | 0.97 | 0.89 |
| ResNet-50x64 | CLIP | OpenAI | 0.83 | 0.27 | 0.53 | 0.99 | 0.60 | 0.85 | 0.99 | 0.95 |
| SwinV2-Base | MiDaS | MiDaS | 0.54 | 0.13 | 0.17 | 0.70 | 0.36 | 0.72 | 0.97 | 0.90 |
| SwinV2-Base-w16 | Supervised | ImageNet-21k-1k | 0.89 | 0.4 | 0.51 | 0.93 | 0.51 | 0.87 | 0.99 | 0.92 |
| SwinV2-Base-w24 | Supervised | ImageNet-21k-1k | 0.89 | 0.43 | 0.51 | 0.94 | 0.52 | 0.87 | 0.99 | 0.93 |
| SwinV2-Large-w16 | Supervised | ImageNet-21k-1k | 0.88 | 0.39 | 0.47 | 0.94 | 0.52 | 0.84 | 0.98 | 0.93 |
| SwinV2-Large-w24 | Supervised | ImageNet-21k-1k | 0.88 | 0.42 | 0.49 | 0.94 | 0.52 | 0.84 | 0.98 | 0.94 |
| SwinV2-Tiny | MiDaS | MiDaS | 0.51 | 0.09 | 0.16 | 0.64 | 0.35 | 0.58 | 0.96 | 0.88 |
| | Supervised | ImageNet-1k | 0.68 | 0.17 | 0.32 | 0.9 | 0.4 | 0.67 | 0.99 | 0.89 |
| ViT-Base | CLIP | LAION-2B | 0.82 | 0.23 | 0.46 | 0.98 | 0.55 | 0.87 | 0.97 | 0.93 |
| | MAE | ImageNet-1k | 0.14 | 0.01 | 0.04 | 0.34 | 0.07 | 0.33 | 0.81 | 0.61 |
| | MoCoV3 | ImageNet-1k | 0.57 | 0.13 | 0.2 | 0.84 | 0.38 | 0.78 | 0.95 | 0.91 |
| | Supervised | ImageNet-1k | 0.56 | 0.12 | 0.23 | 0.84 | 0.3 | 0.56 | 0.95 | 0.86 |
| | Supervised | ImageNet-21k-1k | 0.85 | 0.29 | 0.33 | 0.88 | 0.41 | 0.74 | 0.96 | 0.9 |
| | DINO | ImageNet-1k | 0.72 | 0.21 | 0.29 | 0.91 | 0.43 | 0.85 | 0.99 | 0.91 |
| ViT-Large | MAE | ImageNet-1k | 0.24 | 0.04 | 0.05 | 0.56 | 0.16 | 0.53 | 0.92 | 0.81 |
| | CLIP | LAION-2B | 0.85 | 0.26 | 0.47 | 0.98 | 0.58 | 0.82 | 0.97 | 0.94 |
| | Supervised | ImageNet-21k-1k | 0.88 | 0.38 | 0.47 | 0.9 | 0.47 | 0.83 | 0.97 | 0.92 |
| ViT-Small | MoCoV3 | ImageNet-1k | 0.57 | 0.12 | 0.18 | 0.82 | 0.36 | 0.7 | 0.97 | 0.89 |
| | Supervised | ImageNet-1k | 0.61 | 0.13 | 0.25 | 0.85 | 0.32 | 0.61 | 0.95 | 0.87 |
| | Supervised | ImageNet-21k-1k | 0.84 | 0.27 | 0.3 | 0.88 | 0.43 | 0.74 | 0.96 | 0.88 |
| | DINO | ImageNet-1k | 0.79 | 0.21 | 0.27 | 0.9 | 0.42 | 0.87 | 0.99 | 0.9 |

training period is enough for finetuning, where we only train the model for 100 epochs which is 1/3 as many epochs as we use for training from scratch. Also on smaller datasets, such as Flowers-102 and aircraft datasets, finetuning obtains much higher accuracy compared to training from scratch. In contrast, finetuning does not save quite as many epochs for detection and segmentation where detection systems contain lots of new parameters that are randomly initialized for downstream training.

Table 18: **Recall@5 scores for image retrieval experiments.**

| Backbone | Method | Pretrain Data | CUB | iNat | Obj | INSTRE | GLM | rOxf | rPar | CopyD |
|---|---|---|---|---|---|---|---|---|---|---|
| ConvNext-Base | Supervised | ImageNet-1k | 0.045 | 0.069 | 0.03 | 0.097 | 0.099 | 0.087 | 0.026 | 0.854 |
| | Supervised | ImageNet-21k-1k | 0.085 | 0.221 | 0.046 | 0.111 | 0.163 | 0.174 | 0.027 | 0.902 |
| ConvNext-Small | Supervised | ImageNet-1k | 0.05 | 0.069 | 0.029 | 0.094 | 0.099 | 0.092 | 0.026 | 0.851 |
| | Supervised | ImageNet-21k-1k | 0.084 | 0.196 | 0.042 | 0.106 | 0.151 | 0.169 | 0.027 | 0.888 |
| ConvNext-XLarge | Supervised | ImageNet-21k-1k | 0.086 | 0.247 | 0.048 | 0.111 | 0.164 | 0.172 | 0.027 | 0.903 |
| ResNet-101 | CLIP | OpenAI | 0.05 | 0.06 | 0.02 | 0.106 | 0.117 | 0.082 | 0.026 | 0.848 |
| | Supervised | ImageNet-1k | 0.049 | 0.07 | 0.026 | 0.092 | 0.107 | 0.085 | 0.026 | 0.857 |
| ResNet-50 | CLIP | OpenAI | 0.04 | 0.046 | 0.015 | 0.097 | 0.107 | 0.092 | 0.026 | 0.852 |
| | Supervised | ImageNet-1k | 0.043 | 0.056 | 0.018 | 0.089 | 0.099 | 0.084 | 0.026 | 0.836 |
| | VicReg | ImageNet-1k | 0.026 | 0.042 | 0.009 | 0.09 | 0.116 | 0.106 | 0.027 | 0.868 |
| ResNet-50x64 | CLIP | OpenAI | 0.073 | 0.124 | 0.044 | 0.122 | 0.191 | 0.181 | 0.027 | 0.948 |
| SwinV2-Base | MiDaS | MiDaS | 0.032 | 0.051 | 0.008 | 0.063 | 0.095 | 0.113 | 0.027 | 0.890 |
| SwinV2-Base-w16 | Supervised | ImageNet-21k-1k | 0.083 | 0.207 | 0.043 | 0.109 | 0.154 | 0.178 | 0.027 | 0.917 |
| SwinV2-Base-w24 | Supervised | ImageNet-21k-1k | 0.084 | 0.226 | 0.045 | 0.111 | 0.158 | 0.177 | 0.027 | 0.923 |
| SwinV2-Large-w16 | Supervised | ImageNet-21k-1k | 0.082 | 0.204 | 0.04 | 0.112 | 0.16 | 0.167 | 0.027 | 0.93 |
| SwinV2-Large-w24 | Supervised | ImageNet-21k-1k | 0.083 | 0.223 | 0.041 | 0.113 | 0.161 | 0.17 | 0.027 | 0.931 |
| SwinV2-Tiny | MiDaS | MiDaS | 0.031 | 0.034 | 0.008 | 0.055 | 0.095 | 0.059 | 0.026 | 0.849 |
| | Supervised | ImageNet-1k | 0.052 | 0.072 | 0.021 | 0.101 | 0.114 | 0.096 | 0.027 | 0.868 |
| ViT-Base | CLIP | LAION-2B | 0.071 | 0.104 | 0.037 | 0.12 | 0.172 | 0.175 | 0.027 | 0.92 |
| | MAE | ImageNet-1k | 0.005 | 0.003 | 0.001 | 0.02 | 0.012 | 0.02 | 0.015 | 0.542 |
| | MoCoV3 | ImageNet-1k | 0.036 | 0.055 | 0.01 | 0.088 | 0.1 | 0.114 | 0.026 | 0.887 |
| | Supervised | ImageNet-1k | 0.039 | 0.049 | 0.015 | 0.088 | 0.078 | 0.053 | 0.025 | 0.82 |
| | Supervised | ImageNet-21k-1k | 0.077 | 0.146 | 0.024 | 0.094 | 0.111 | 0.088 | 0.026 | 0.883 |
| | DINO | ImageNet-1k | 0.055 | 0.093 | 0.018 | 0.102 | 0.125 | 0.158 | 0.027 | 0.894 |
| ViT-Large | CLIP | LAION-2B | 0.077 | 0.122 | 0.038 | 0.12 | 0.183 | 0.127 | 0.027 | 0.934 |
| | MAE | ImageNet-1k | 0.01 | 0.012 | 0.002 | 0.042 | 0.036 | 0.038 | 0.023 | 0.753 |
| | Supervised | ImageNet-21k-1k | 0.082 | 0.207 | 0.04 | 0.099 | 0.141 | 0.15 | 0.027 | 0.9 |
| Vit-Small | MoCoV3 | ImageNet-1k | 0.037 | 0.05 | 0.009 | 0.082 | 0.097 | 0.094 | 0.026 | 0.877 |
| | Supervised | ImageNet-1k | 0.044 | 0.053 | 0.016 | 0.088 | 0.085 | 0.062 | 0.025 | 0.843 |
| | Supervised | ImageNet-21k-1k | 0.076 | 0.126 | 0.021 | 0.097 | 0.118 | 0.097 | 0.026 | 0.869 |
| | DINO | ImageNet-1k | 0.065 | 0.093 | 0.016 | 0.101 | 0.123 | 0.16 | 0.027 | 0.888 |

Table 19: **Top-1 classification accuracy (%) for ImageNet against adversarial attacks with $\ell_\infty$ constraint radii** $1/255$ **and** $2/255$.

| Backbone | Method | Pretrain Data | Clean | $\epsilon = 1/255$ | $\epsilon = 2/255$ | Z-scores |
|---|---|---|---|---|---|---|
| ResNet-50 | Supervised | ImageNet-1k | 80.38 | 28.79 | 13.25 | -0.99 |
| | VicReg | ImageNet-1k | 78.77 | 36.60 | 22.01 | -0.26 |
| | CLIP | LAION-2B | 80.55 | 32.78 | 18.55 | 0.58 |
| | DINO | ImageNet-1k | 79.08 | 35.80 | 20.75 | -0.35 |
| ResNet-101 | Supervised | ImageNet-1k | 81.93 | 44.23 | 32.43 | 0.53 |
| ViT-Small | Supervised | ImageNet-1k | 78.84 | 21.21 | 6.42 | -1.62 |
| | Supervised | ImageNet-21k | 81.39 | 16.50 | 3.83 | -1.94 |
| | MoCoV3 | ImageNet-1k | 79.65 | 48.62 | 32.16 | 0.71 |
| | DINO | ImageNet-1k | 81.33 | 47.87 | 30.04 | 0.57 |
| ViT-Base | Supervised | ImageNet-1k | 79.15 | 27.08 | 9.54 | -1.23 |
| | Supervised | ImageNet-21k | 84.53 | 23.04 | 6.92 | -1.52 |
| | MoCoV3 | ImageNet-1k | 82.85 | 55.44 | 39.49 | 1.32 |
| | MAE | ImageNet-1k | 83.41 | 50.85 | 31.69 | 0.77 |
| | DINO | ImageNet-1k | 83.40 | 53.59 | 36.61 | 1.11 |
| | CLIP | LAION-2B | 85.19 | 47.91 | 28.35 | 0.50 |
| SwinV2-Tiny | Supervised | ImageNet-1k | 81.82 | 40.91 | 23.15 | -0.03 |
| | MiDaS | MiDaS | 82.55 | 41.44 | 25.20 | 0.08 |
| ConvNeXt-Tiny | Supervised | ImageNet-1k | 82.10 | 49.74 | 31.42 | 0.72 |
| ConvNeXt-Base | Supervised | ImageNet-1k | 83.88 | 55.31 | 37.19 | 1.21 |
| | Supervised | ImageNet-21k | 85.87 | 53.78 | 34.05 | 1.00 |

Table 20: **Z-scores for best-performing SSL and supervised learning backbones.** Mean z-scores for each task averaged across the 3 top performing backbones dividing models into self (weakly-)-supervised learning (SSL) on ImageNet-1k, supervised learning on ImageNet-1k (Sup-1k), and ImageNet-21k (Sup-21k).

| Task | SSL | Sup-1k | Sup-21k |
|---|---|---|---|
| Cls | 0.573 | 0.527 | 0.936 |
| Det | 0.298 | 0.743 | 1.076 |
| Seg | 0.314 | 0.717 | 1.071 |
| Ret | 0.489 | -0.079 | 0.708 |
| (OOD) Cls | 0.419 | 0.287 | 1.271 |
| (OOD) Det | 0.414 | 0.923 | 0.853 |

