# OpenReview forum: "Battle of the Backbones: A Large-Scale Comparison of Pretrained Models across Computer Vision Tasks"
_NeurIPS.cc/2023/Track/Datasets_and_Benchmarks — NeurIPS 2023 Datasets and Benchmarks Poster_

### Official Review · Reviewer_eVj3 · 2023-07-11
**A extensive benchmark of computer vision backbones**

**Rating:** 8
**Confidence:** 4
**Correctness:** Claims are supported by the experimen…
**Clarity:** Very well written and the takeaways a…

**Strengths:**

- This paper provides an extensive benchmark of different pretrained models for a series of computer vision tasks. It reveals the strengths and weaknesses of various pretraining routines and architectures, providing valuable insights for practitioners and researchers alike.
- The experiments were conducted with many different datasets for diverse tasks and thus the meta-analysis is more likely to be generalizable.
- The open-source codebase can be a valuable resource for the community.

**Additional Feedback:**

NA

**Documentation:**

The authors have open-sourced code and provided detailed documentation in the supplementary materials.

**Ethics:**

No ethical concerns.

**Limitations:**

As the authors have pointed out, this paper mainly focuses on the performance. But when choosing the model, fairness is another important aspect to consider. For example, different biases introduced by vision-language pertaining (e.g., CLIP) and vision-only pretraining (e.g., ConvNeXt) are not studied.

**Opportunities For Improvement:**

Doesn't have to be a weakness, but:
- The authors may consider some statistical tests for further comparisons. For example, for multiple algorithms on multiple datasets, post-hoc Nemenyi test can be used to show the relative rank and statistical significance. I think the conclusions would be more convincing if there is a statistical significance.
- More pretraining methods can be included.

**Relation To Prior Work:**

The authors have provided comprehensive descriptions of previous works.

**Summary And Contributions:**

This paper proposed the Battle of the Backbones (BoB), a comprehensive benchmark that compares various pretrained model backbones across a range of computer vision tasks. It provides valuable insights for practitioners and researchers and discusses directions for future computer vision research.

---

> ### Author Response · Authors · 2023-08-23
> **Thank you for your feedback**
>
> Thank you for your supportive and constructive feedback.  Our work provides a thorough comparison of pretrained models and their various properties that a vast crowd of practitioners can employ when choosing a backbone.  Moreover, our work is informative for researchers interested in the strengths and weaknesses of existing approaches to pretraining.
>
> Regarding statistical significance, we will perform the post-hoc Nemenyi test and also include p-values for all correlations measured in our camera ready version.  Interestingly, our study of the relationship between performance and scale (training data size, number of parameters, etc.) shows that transformers benefit especially from scale, and the Spearman rank correlations in these cases are accompanied by very small p-values, demonstrating the extent that transformers benefit from scale.  We will discuss this further in our camera ready version.  Moreover, we release all numerical results so that any researcher or practitioner can apply their own analysis subsequently.
>
> We have additionally now benchmarked Segment Anything Model (SAM) ViTs, which were pretrained on 11 million images accompanied by over a billion masks, and will include them in our camera ready version.  We found that these models outperform all other ViT backbones from our benchmark, but they still underperform more modern architectures such as ConvNeXt and SwinV2.  We have also included very small models including EfficientNet and RegNet, which are used in lightweight practical applications (see Section 4.3 from our updated draft for very small models). We also point out that our benchmark will be open source and evolving so that new pre-trained backbones can be put through the gauntlet as they come out.
>
> We agree that fairness is also of practical importance for many applications.  Among vision applications, a task where fairness is of utmost importance is face recognition, however the pre-trained backbones we consider are not competitive for face recognition since large-scale face rec datasets exist.  In the future, we will consider incorporating Dollar Street, a natural image dataset with proper annotations for computing fairness metrics.

---

### Official Review · Reviewer_YjWA · 2023-07-18
**A comparison of a variety of pretrained models (i.e., backbones) across computer vision tasks**

**Rating:** 6
**Confidence:** 4

**Strengths:**

1) Transfer learning from the pretrained backbones has emerged as an important topic in computer vision and language tasks. The paper handles representative backbone architectures with recently popular pretraining methods. Their pretrained checkpoints are available in public, which provides high accessibility to practitioners.

2) The clear and meaningful message is given: among backbones evaluated in this benchmark study, three models, named Sup. ConvNeXt-B (In-21k), Sup. SwinV2-B (IN-21k,1k), and CLIP ViT-B (LAION-2B), are recommended to be utilized by practitioners and researchers in 2D computer vision tasks, as a backbone.

3) Through the extensive experiments in the four computer vision tasks, the authors provide a unified conclusion in the backbone selection. This may have an impact to practitioners who consider backbones with model parameters of ~150M or less.

**Additional Feedback:**

  A. How many epochs are used in linear probing?

  B. Why are the MiDaS-based backbones (SwinV2) excluded from Table 2 in the main paper? What factors of fairness do authors consider?

  C. Sup. SwinV2-B (IN-21k,1k), which is the best backbone of the classification task in Table 2 in the main paper, does not exist in Table 6 and 7 in Appendix. Is it a mistake?

  D. There may be an inconsistent point that Swin2-Base-Supervised-IN-1k in Table 6 in Appendix and Swin2-Base-Supervise-IN-21k in Table 7 in Appendix have the same ImageNet accuracy, 87.10.

  E. As a person familiar with CIFAR100, the results of backbones trained from scratch (denoted by Rand Init in Method in Table 7) are much lower, meaning that the used training scheme seems to be inappropriate. Could the authors explain these results? In addition, it is recommended to check the results of the other datasets when using Rand Init.

  F. What is AP^box? It is different from mAP@50? To ensure clarity for readers and to prevent misunderstandings, this paper should include self-contented explanations of the metrics used in object detection and segmentation tasks.

  G. In Tables 8, 9, and 11 in Appendix, do 12-depths of backbones with MiDaS mean using 12 datasets within MiDaS dataset?

  H. In Tables 12, 13, and 14 in Appendix, is IN-22k a typo of IN-21k?

  I. In some configs, whose suffix is “_ft” (which maybe used for finetuning in this benchmark), located at github, “pretrained” option is set to False. Is it right? Or just typo?

**Clarity:**

Overall, the logic is well-written. But, it is highly recommended that the paper be revised, taking into account the feedback mentioned in the "Opportunities for Improvement".

**Correctness:**

Overall the claims made in the submission seems to be correct. The evaluation is conducted fairly for each backbone as much as possible.

However, due to the limited range of model sizes, some of the claims might be difficult to say completely correct in general (e.g., comparison of architectural priors: ViTs vs CNNs)

**Documentation:**

The authors provide sufficient detail to support reproducibility.

**Ethics:**

Irrelevant

**Limitations:**

Yes, the authors write the limitations and potential negative societal impact such as exploring biases in models.

**Opportunities For Improvement:**

1) As shown in Figure 2, while CNNs (ResNet, ConvNeXt) are favored to employ linear probing strategy, end-to-end finetuning is more effective for Transformer-based models (ViT, SwinV2). It is valuable to clarify (or discuss at least) the reason why such different aspect was observed.

2) The aim to reduce choices about backbone selection for practitioners and researchers is commendable. However, behind just recommending top-tier backbones (harshly speaking, this is quite trivial because researchers can obtain clues from other studies using these backbones), it would be valuable to provide a detailed description of finetuning strategies, focusing on the specific modifications required in contrast to training from scratch; for example, compared to training from scratch, using a lower learning rate (how lower?), conducting a shorter training period (how shorter?), and so on.

3) In the similar reason, for Tables 6, 8, and 9 in Appendix, reporting the computation cost (which can be measured by GPUhours or GPUdays) required by end-to-end finetuning and linear probing would be helpful to determine which strategy the practitioners and researchers employ in their CV task.

4) It would be better to improve the readability of tables in Appendix and use a common format across all the tables. As authors know, different formats can be seen, where the contents may be difficult to understand: case 1) all descriptions are included in “Backbone”, case 2) dividing them into columns named Backbone-method-dataset, and case 3) columns named Training-Architecture.

5) Even though the authors mentioned that they do not consider architectures bigger than ConvNeXt-Base, it could be necessary to take into account bigger ones with billion units of model parameters, since they are getting more and more attention as the state-of-the-arts.

**Relation To Prior Work:**

The paper clearly discusses how this work differs from previous contributions.

**Summary And Contributions:**

Authors thoroughly conducted experiments that finetune (or do linear-probing) and evaluate neural networks used as a backbone in the 2D computer vision (CV) field using five tasks (classification, detection, segmentation, out-of-distribution generalization, retrieval) with various datasets. These backbone networks were trained by supervised learning, semi-supervised learning, contrastive learning, and a scheme used for image generation. Among them, supervised ConvNeXt-Based, supervised SwinV2-Base trained with ImageNet-21K and ViT-Base trained with CLIP method can be the best backbone used for various CV tasks.

---

> ### Author Response · Authors · 2023-08-23
> **Thank you for your feedback (response 1/2)**
>
> Thank you for your supportive and constructive feedback.  Our work provides a thorough comparison of pretrained models and their various properties that a vast crowd of practitioners can employ when choosing a backbone.  Moreover, our work is informative for researchers interested in the strengths and weaknesses of existing approaches to pretraining.  We respond to each of your points below:
>
> 1.  The observation that transformers benefit more from end-to-end fine-tuning than convolutional backbones, which we make in Section 5, is interesting.  The shortcomings of transformers for linear probing has been observed previously but for specific pretrained models (e.g. He et al. “Masked autoencoders are scalable vision learners”), and we verify this trend at a larger scale.  Other work has found that vision transformers trained with CLS tokens contain more localization in their very deep features which is unnecessary for classification but may be useful for dense prediction tasks (Raghu et al. “Do Vision Transformers See Like Convolutional Neural Networks?”). We want to be cautious speculating about why this trend may occur, but one reason that would be consistent with Raghu et al. above is that the intrinsic dimension of transformer features may be higher than that of CNN features, a trait which has been shown damaging to sample complexity and linear probing.  Investigating this difference between architectural styles is a promising direction for future work.
> 2.  For hyperparameter tuning, we find that the learning rate strategy is highly method- and dataset-dependent. For example, on ImageNet classification, the best learning rate we tried for CLIP was 1e-4 while the best learning rate for MAE was 1e-3, which is similar to the best learning fate for training from scratch. We speculate that this learning rate sensitivity occurs because different pretraining algorithms lead to parameter vectors of very different magnitudes.  For image classification, a shorter training period is enough for finetuning, where we only train the model for 100 epochs which is 1/3 as many epochs as we use for training from scratch. Also on smaller datasets, such as Flowers-102 and aircraft datasets, finetuning obtains much higher accuracy compared to training from scratch. In contrast, finetuning does not save quite as many epochs for detection and segmentation where detection systems contain lots of new parameters that are randomly initialized for downstream training.  We have now included these high-level observations in our updated draft.
> 3.  We agree that knowing the computational tradeoff would be valuable for practitioners.  We now evaluate the training costs on ImageNet for end-to-end finetuning and linear probing on 8 NVIDIA A5000 GPUs. Among all the backbones, on average, end-to-end finetuning takes 12.6 minutes per epoch whereas linear probing takes 10.4 minutes. Notably, for small networks that require less memory, such as ResNet-50 and ViT-Small, the time spent processing data dominates the training time, which makes the end-to-end training and linear probing speeds similar. Therefore, we also collect the training time for large models only including SwinV2-base variants, ConvNeXt-Base, and the Stable Diffusion encoder. On average for these large models, end-to-end finetuning takes 27.8 minutes per epoch and linear probing takes only 13.1 minutes.  We will add this discussion to the camera ready version, and we will add the raw training times to the benchmark github before we release it.
> 4.  Thanks for pointing out our inconsistent formatting.  We have now corrected this inconsistency in our updated draft and chosen a format that we think is particularly interpretable.
> 5.  You are correct that some modern vision backbones, including state-of-the-art on some tasks, are very large.  Adding large backbones pretrained using various pretraining algorithms and benchmarking them on our diverse suite of tasks is computationally infeasible for us.  In contrast, Reviewer fJFu pointed out that we already benchmarked models which are too large to be practically useful for many applications, so we have added very small models including EfficientNet-B0 and RegNetX-400F (see Section 4.3 from our updated draft.  For many practitioners, models which are so large that they must be sharded or even parallelized across nodes may be impractical.  Nonetheless, our open-source benchmark will be evolving, so we hope researchers will add very large models to our benchmark in the future to compete.

---

> > ### Author Response · Authors · 2023-08-23
> > **Thank you for your feedback (response 2/2)**
> >
> > A.  90 epochs
> > B.  We excluded MiDaS backbones from ImageNet comparisons because they were actually pretrained on ImageNet prior to being fine-tuned on depth tasks.
> > C.  Thanks for catching this mistake.  The model was actually in both places, but we accidentally used inconsistent names.  We have fixed this in our updated draft.
> > D.  Thank you for pointing out this typo.  Both of those should say 21k.  We have corrected this in our updated draft.
> > E.  Thanks for bringing this issue to our attention. We have now run additional experiments with a larger learning rate and more iterations, specifically for models trained from scratch. Our latest results show that a random initialized ResNet-50 trained on CIFAR-100 for 100 epochs obtains top-1 accuracy of 72.33%, and similar improvements on other architectures. In addition, we also re-ran the updated grid search on other datasets and observed similar improvements in accuracy. We have now updated the results in the associated table in our updated draft.
> > F.  We follow the COCO-style average precision (AP) metric, which calculates the average across various Intersection over Union (IoU) thresholds (please refer to the Lin et al. “COCO Detection Evaluation” for more details). To enhance comparability and clarity, we have now added AP^box @50 and AP^box @75 to our evaluation metrics (see Tables 8 and 9 in our updated draft).  We also have added a description of the AP metric and a reference to the associated paper (see Section 3.2).
> > G.  “12-depths” refers to the MiDaS dataset which combines 12 different depth datasets.  We realize that this was confusing, so we have now removed this from our updated draft since all MiDaS backbones we include are pretrained on the same dataset which we describe earlier in the paper.
> > H.  We have now corrected these tables, which should all say IN-21k.
> > I.  In our launch scripts, we overwrite hyper-parameters for grid search including `pretrained`. Specifically, we overwrite `pretrained` to be `True` for fine-tuning experiments, so this is indeed a typo we missed while cleaning up our code for submission but one that does not effect experiments. We will update the code repository accordingly before public release. Thank you for noticing this oversight.
> >
> > We have additionally now benchmarked Segment Anything Model (SAM) ViTs, which were pretrained on 11 million images accompanied by over a billion masks, and will include them in our camera ready version. We found that these models outperform all other ViT backbones from our benchmark, but they still underperform more modern architectures such as ConvNeXt and SwinV2.
> >
> > Thank you again for your thoughtful review. We made a significant effort to address your feedback including new experiments and paper edits, and we would appreciate it if you would consider raising your score in light of our response.  Please let us know if you have additional questions we can address.

---

> > > ### Comment · Reviewer_YjWA · 2023-08-25
> > >
> > > Thank the authors for the detailed answers. Many of the concerns have been resolved.
> > > Nevertheless, I still have the following questions.
> > >
> > > 1. I fully understand the 5-th comment in the author response.
> > > Nevertheless the reason I thought consideration of bigger models is necessary is as follows.
> > > In the constrained experimental settings, the authors insist that "Supervised ConvNeXt" is the best choice of backbones under model parameters of ~150M or less. However, it might be doubtful that the claim is still valid in general (e.g., bigger size of architectures, multi-stage pretraining, and so on).
> > > Since in the paper the authors reported that the correlation of task performance with increasing model size is higher for ViTs than CNNs, there seems to have possibility that ViTs outperform CNNs in bigger models (e.g., with billion units of parameters).
> > > Hence, I still feel that it is necessary to discuss "the generality of the claim"; more empirical evidence or theoretical/logical analysis would be welcomed.
> > >
> > > I appreciate again the thorough response.

---

> > > > ### Author Response · Authors · 2023-08-25
> > > > **Thank you for your feedback**
> > > >
> > > > We agree with your point, and we have now updated our draft to clarify.  We have added an additional paragraph on this limitation in Section 6.  Further, we have added several sentences throughout the paper qualifying relevant conclusions, for example pointing out in our discussion of CNNs vs ViTs that the greater effect size of scale for ViTs might indicate that transformers may outperform convnets at much larger scales.  Including very large backbones in our benchmark is computationally infeasible for us, but we appreciate this limitation and we welcome creators of new backbones to add their model to our open-source public benchmark.  We once again thank you for your feedback.  Please let us know if you have any additional questions we can address that could improve our work or assist you in your assessment.

---

### Official Review · Reviewer_fJFu · 2023-07-18

**Rating:** 6
**Confidence:** 4
**Clarity:** The writting of this paper is good, a…

**Strengths:**

1. The paper is well-written with a clear organization, and the authors provide very detailed implementation specifics.
2. The authors conduct comprehensive experiments to compare the performances of various backbones across different downstream tasks. To ensure a fair comparison, they meticulously adjust hyperparameters in downstream tasks to minimize their influence.
3. The motivation for this study is compelling, and I am convinced that this benchmark can serve as a valuable empirical reference for the community.

**Additional Feedback:**

Please refer to Opportunities For Improvement

**Correctness:**

This paper propose a benchmark and the evaluation methods and experiment design are correct.

**Documentation:**

This paper describe detailed implementation specifics in supplementary material and I believe it is sufficient to support reproducibility.

**Ethics:**

I consider there are not ethical concerns.

**Limitations:**

The authors partially addressed the comparison of smaller models by providing an analysis for models with <30M parameters. However, there is still a lack of comparisons for even smaller models that are more commonly used in practical applications.

**Opportunities For Improvement:**

1.In practical applications, such as autonomous driving scenarios, users have high demands for the computational efficiency of the model. This paper primarily focuses on performance comparisons of some large-parameter backbones, lacking discussions on lightweight models. Even though the paper mentioned comparisons of models with <30M parameters, actual users may pay more attention to models with <10M parameters, such as ResNet-18, EfficientNet-B0 (B1, B2), or RegNet-400MF (600MF, 800MF), etc.
2. This paper lacks comparisons of some classic convolutional neural networks, such as EfficientNet and RegNet. The authors only discussed ResNet and ConvNeXt. Please explain the reason for this.
3. Some of the experiments in the paper, such as Detection training, adopt a 3x training strategy. The results of MAE ViT-B are lower compared to Sup. SwinV2-B (IN-21k). However, in the VitDet paper, the results of VitDet-B are roughly on par with Swin-B (21K, sup). VitDet-L even exceeds Swin-L. Is the conclusion that "Supervised ConvNeXt-Base trained on IN-21K" > "Supervised SwinV2-Base trained on IN-21k (finetuned on IN-1k)" > "Supervised ConvNeXt-Base trained on IN-1k" solid enough based on these observations?

**Relation To Prior Work:**

The authors discuss the prior works clearly in Sec. 1.2 and Appendix 2.

**Summary And Contributions:**

This paper builds a benchmark named Battle of the Backbones (BoB) to evaluate the performances of different backbones (with different pre-training paradigms) across various tasks, including image classification, object detection and segmentation, out-of-distribution generalization, and image retrieval. According to the benchmark results, the authors provide several observations and guidelines, which may assist users in choosing suitable backbones for practical usage.

---

> ### Author Response · Authors · 2023-08-23
> **Thank you for your feedback**
>
> Thank you for your supportive and constructive feedback.  Our work provides a thorough comparison of pretrained models and their various properties that a vast crowd of practitioners can employ when choosing a backbone.  Moreover, our work is informative for researchers interested in the strengths and weaknesses of existing approaches to pretraining.  We respond to each of your points below:
>
> Prompted by your feedback regarding classic and small architectures, we have now added experiments with ResNet-18, RegNet-400MF, and EfficientNet-B0. We have now added these experiments to our updated draft, Section 4.3.
>
> In our experimental setup, we prioritize maintaining fair comparisons among different backbones, instead of using higher performing pipelines for some backbones than others. Consequently, we aimed to minimize architectural modifications to our detectors. We thus avoid implementing some ViTDet modifications such as “simple feature pyramid” instead of FPN. Therefore, for our ViT-based detectors, we employ an FPN that utilizes the final feature map, without employing stage division. It’s worth noting, as highlighted in the ViTDet paper, the performance of “FPN, last-map” is inferior to the “simple feature pyramid” (see Table 1, rows b, c in [A]). We have now clarified these details in our updated draft to avoid confusion.
>
> We have additionally now benchmarked Segment Anything Model (SAM) ViTs, which were pretrained on 11 million images accompanied by over a billion masks, and will include them in our camera ready version. We found that these models outperform all other ViT backbones from our benchmark, but they still underperform more modern architectures such as ConvNeXt and SwinV2.
>
> Thank you again for your thoughtful review. We made a significant effort to address your feedback including new experiments and would appreciate it if you would consider raising your score in light of our response.  Please let us know if you have additional questions we can address.
>
> [A] - Exploring Plain Vision Transformer Backbones for Object Detection

---

> > ### Comment · Reviewer_fJFu · 2023-08-29
> >
> > Thanks for your response. I consider maintain my oringinal score. I still consider the conclusion of Object Detection is not solid enough.

---

> > > ### Author Response · Authors · 2023-08-29
> > > **Thank you for your feedback**
> > >
> > > Thank you again for your feedback.  We agree that bigger architectural modifications and very long training schedules can benefit ViTs in particular, as was found in the ViTDet paper.  Similarly, [1] point out that ViTDet achieves stronger performance than their own work due to long and expensive training routines, behavior which stems from ViTs weak vision inductive bias. We also want to point out that our analysis indicates that ViTs benefit more from scale, so it might be the case that ViTs will overtake other models at larger scales.  Prompted by your feedback, we have now added a discussion about these observations to our new updated draft.  Including large models in our benchmark, which includes many tasks and pretraining methods, would be prohibitively expensive and as you point out, they are impractical for many real-world applications.  Moreover, our inclusion of LVIS and Sim10k→Cityscapes enhances the rigor of our object detection findings.  We appreciate your point though and agree that including this disclaimer will be useful for practitioners with the resources to train large models, especially with long training routines and architectural modifications which incorporate additional parameters.  Please let us know if you have any additional questions.
> > >
> > > **References**
> > > [1] Chen, Zhe, et al. "Vision Transformer Adapter for Dense Predictions." The Eleventh International Conference on Learning Representations. 2023.

---

### Official Review · Reviewer_SBvp · 2023-07-21
**Paper Review**

**Rating:** 6
**Confidence:** 5
**Correctness:** Yes.
**Clarity:** Yes.

**Strengths:**

- It is easy to follow.
- It can be useful information when selecting backbone models for some parts of computer vision tasks.
- Extensive experiments.

**Additional Feedback:**

Please resolve my concerns.

**Documentation:**

Yes.

**Ethics:**

No.

**Limitations:**

- This paper provides good information about comparing backbone models. However, it fails to explain why Sup. ConvNeXt is best in Det, Seg, Ret, and OOD or Sup. Swin V2-B is the best in Cls. Extensive experiments are a good trial, but I think many researchers or practitioners want to know why the backbone models are good for the task. If this is included, it will be a better paper.

- The authors selected several tasks for experiments, such as classification, object detection & segmentation, image retrieval, and OOD tasks. Why did you select only those tasks? There are many other tasks in computer vision, such as human pose estimation, depth estimation, super-resolution, deblurring, and so on. What about these tasks? Would ConvNext and Swin also be the best?

- For the classification task, several datasets are used for experiments Furthermore, medical and satellite datasets are used which implies a large domain gap between pre trained dataset and the target dataset. However, only COCO dataset is used for object detection and segmentation tasks. Similar to the classification experiments, how about adding large domain gap datasets of medical or satellite for object detection and segmentation tasks? (medical: [1], satellite: [2], [3])

[1] https://paperswithcode.com/datasets?mod=medical

[2] Ke Li, Gang Wan, Gong Cheng, Liqiu Meng, and Junwei Han. Object detection in optical remote sensing images: A survey and a new benchmark. ISPRS Journal of Photogrammetry and Remote Sensing, 159:296–307, 2020

[3] Jian Ding, Nan Xue, Gui-Song Xia, Xiang Bai, Wen Yang, Michael Yang, Serge Belongie, Jiebo Luo, Mihai Datcu, Marcello Pelillo, and Liangpei Zhang. Object detection in aerial images: A large-scale benchmark and challenges. IEEE Transactions on Pattern Analysis and Machine Intelligence, pages 1–1, 2021.

**Opportunities For Improvement:**

See Limitations.

**Relation To Prior Work:**

Yes.

**Summary And Contributions:**

Most deep learning algorithms for computer vision systems utilize backbone models. The selection of a backbone model is important for achieving better performance in classification, object detection and segmentation, out-of-distribution, and image retrieval. This paper provides performed extensive experiments to reveal which backbone is useful for each task. Finally, the paper found ConvNeXt and Swin V2-B models are best in classification and other tasks, respectively. The one of merits of the paper is that it can be a good guide for practitioners.

---

> ### Author Response · Authors · 2023-08-23
> **Thank you for your feedback**
>
> Thank you for your supportive and constructive feedback.  Our work provides a thorough comparison of pretrained models and their various properties that a vast crowd of practitioners can employ when choosing a backbone.  Moreover, our work is informative for researchers interested in the strengths and weaknesses of existing approaches to pretraining.  We respond to each of your points below.
>
> Our work reports differences in behavior between SSL and supervised pretraining as well as differences in behavior between convolutional and transformer architectures.  We found it surprising that supervised ConvNeXt exhibits such broadly strong performance compared to recent state-of-the-art self-supervised learning approaches.  Several other papers do attempt an understanding of how different architectures or pretraining strategies work, for example transformers vs. convolutional networks [1] and supervised vs. self-supervised learning [2-3], and we agree that further work on understanding the structural differences between backbones that causes the performance differences is a promising direction for future work.
>
> Due to the already high computational costs of this benchmark, we chose a diverse cross-section of representative computer vision tasks ranging from classification to dense prediction (detection and segmentation) and to OOD generalization.  Nonetheless, we agree that adding additional computer vision tasks would provide value to practitioners who work on those tasks, so we encourage the addition of such tasks to our open-source benchmark, and we hope to add more tasks ourselves in the future, including as new exciting datasets emerge.
>
> We wanted to include diverse downstream domains, so we ran experiments in medical and satellite imaging domains for classification where computational costs were feasible.  Note that we also included Sim10k→Cityscapes object detection which differs strongly from COCO in nature.  We have now run preliminary experiments on LVIS, which contains more than 1200 categories and images in diverse settings, as we describe in our response to Reviewer eLJh, but we do agree that medical and satellite imaging experiments for dense prediction tasks would be worthwhile, and we encourage other interested researchers from these domains to add these to our open-sourced benchmark.
>
> We have additionally now benchmarked Segment Anything Model (SAM) ViTs, which were pretrained on 11 million images accompanied by over a billion masks, and will include them in our camera ready version.  We found that these models outperform all other ViT backbones from our benchmark, but they still underperform more modern architectures such as ConvNeXt and SwinV2.  We have also included very small models including EfficientNet and RegNet, which are used in lightweight practical applications (see Section 4.3 from our updated draft for very small models).
>
> Thank you again for your thoughtful review. We made a significant effort to address your feedback and would appreciate it if you would consider raising your score in light of our response.  Please let us know if you have additional questions we can address.
>
> **References**
> [1] Park, Namuk, and Songkuk Kim. "How do vision transformers work?."
> [2] Shwartz-Ziv, Ravid, et al. "What Do We Maximize in Self-Supervised Learning And Why Does Generalization Emerge?." (2022).
> [3] Ben-Shaul, Ido, et al. "Reverse Engineering Self-Supervised Learning."

---

### Official Review · Reviewer_eLJh · 2023-07-24
**Benchmark of a diverse suite of pretrained models across a diverse set of computer 11 vision tasks**

**Rating:** 5
**Confidence:** 5

**Strengths:**

The paper presents a comprehensive comparative analysis of the most recent and high-performing models in the literature.
The comparison presented confirms existing observations that larger models perform better at tasks and Vits are more sensitive to the amount of training data available, and show that performance across tasks is strongly correlated.
The ablations presented use (pre-existent) publicly available checkpoints that can accessible to practitioners interested follow up with in the models ablated.
These checkpoints cover models pretrained with supervised, vision-language and SSL.
In conclusion, a one-to-one comparison of public checkpoints can be highly beneficial for practitioners who seek off-the-shelf solutions and do not have the resources or interest in developing their own solutions.

**Additional Feedback:**

It is not clear if this can be considered a dataset paper as evaluation datasets (In Table 3, Table 4, and Table 5) are publicly available. Its contribution is the overall comparison of public checkpoints, but it is not clear what would be the 'dataset' contribution. Please clarify.

**Clarity:**

The text, ablations and results are clear and easy to follow. Tables contain most common metrics. The paper and supplemental material contain many pointers to the backbones, datasets and  styles covered.

**Correctness:**

The ablations cover publicly available checkpoints, thus, can be assumed to replicate original models by construction. Experiments are well described and detailed (considering the Sup. material).
Part of the submission (github) contains code for replicating the evaluation, thus it can be used for benchmarking new backbones under same protocol.



**Documentation:**

The github contains supporting code for replicating the evaluations presented, and Appendix F describes Dependencies and Licenses.

**Opportunities For Improvement:**

The major contribution of the paper is to present an independent comparison of public checkpoints. As a consequence of this choice, its major limitation are the absence of independent one-to-one training and also the lack of  explorations covering new combinations (backbone versus dataset versus style) not covered by the corresponding original papers. In this sense, the conclusion and observations replicate the differences and biases from the original pipelines and do not shed a light on how they may alter the cross backbone comparison.

The object detection and segmentation analysis is limited. The paper explores COCO dataset. On the one hand, this is a well-known dataset and it is important to revalidate results on it. However, its correlation with ImageNet classification results is well-known, and more challenging datasets such as LVIS, which provides a large number of categories and where per-category data is sometimes scarce, may be more appropriate for evaluating the performance of new methods and for enhancing the differences of the different backbones.



**Relation To Prior Work:**

The paper does not present a new method but instead focus on an independent comparison of public available backbone and clearly covers important related work on strong backbone and training styles.

**Summary And Contributions:**

A large comparison of pretrained models on visual tasks including classification benchmarks, object detection and segmentation benchmarks, and  image retrieval. The models evaluated represent the major architectures trends including Resnets, ConvNext, Vision transformers, Swin Tranformers and Stable diffusion encoder. The paper points  ConvNeXt-Base and supervised SwinV2-Base trained using ImageNet-21k and CLIP ViT-Base emerge as the best performing models, while their smaller versions also emerge as winners when considering only smaller backbones.
Ablations were made from pre-existent publicly available checkpoints and cover supervised and SSL models tested on iid and ood datasets.

---

> ### Author Response · Authors · 2023-08-23
> **Thank you for your feedback**
>
> Thank you for your supportive and constructive feedback.  Our work provides a thorough comparison of pretrained models and their various properties that a vast crowd of practitioners can employ when choosing a backbone.  Moreover, our work is informative for researchers interested in the strengths and weaknesses of existing approaches to pretraining.  We respond to each of your points below.
>
> We agree that a wider variety of detection and segmentation experiments could provide further analysis of backbone behaviors.  Note that we did also include OOD object detection experiments on the Sim10k→Cityscapes setup.  Moreover, prompted by your suggestion, we have now run experiments on LVIS using the SwinV2-T, SwinV2-B, ConvNeXt-T, and ConvNeXt-B backbones all pretrained in a supervised fashion on ImageNet-1k, as well as MiDaS SwinV2-T.  We are currently running the remaining LVIS experiments, including a more extensive hyperparameter sweep, and will include LVIS results with all backbones in the camera ready version.  So far, we see from our preliminary results that the model comparisons are similar, except that MiDaS is a bit worse compared to others than previously, and the performance gaps are also smaller.  We agree that these experiments improve the thoroughness of our benchmark, and we include preliminary results below:
> ​
> | Backbone       | Method     | Data        | Params | Input Size | $\text{AP}^\text{box}$ | $\text{AP}^\text{box}_{50}$ | $\text{AP}^\text{box}_{75}$ | $\text{AP}^\text{mask}$ | $\text{AP}^\text{mask}_{50}$ | $\text{AP}^\text{mask}_{75}$ |
> |----------------|------------|-------------|--------|------------|------------------|----------|---------|--------|-----------|-----------|
> | SwinV2-Tiny    | Supervised | ImageNet-1k | 104M   | 1333 × 800 | 33.0             | 46.3     | 35.3    | 29.9   | 44.5      | 31.9      |
> | SwinV2-Tiny    | MiDaS      | MiDaS       | 104M   | 1333 × 800 | 32.6             | 45.7     | 34.9    | 29.6   | 43.9      | 32.0      |
> | ConvNeXt-Tiny  | Supervised | ImageNet-1k | 104M   | 1333 × 800 | 33.2             | 46.1     | 35.4    | 29.9   | 44.3      | 32.2      |
> | SwinV2-Base-w8 | Supervised | ImageNet-1k | 163M   | 1333 × 800 | 35.7             | 48.7     | 38.0    | 32.0   | 46.9      | 34.4      |
> | ConvNeXt-Base  | Supervised | ImageNet-1k | 164M   | 1333 × 800 | 35.8             | 48.8     | 38.0    | 32.0   | 47.0      | 34.5      |
>
> We have additionally now benchmarked Segment Anything Model (SAM) ViTs, which were pretrained on 11 million images accompanied by over a billion masks, and will include them in our camera ready version.  We found that these models outperform all other ViT backbones from our benchmark, but they still underperform more modern architectures such as ConvNeXt and SwinV2.  We have also included very small models including EfficientNet and RegNet, which are used in lightweight practical applications (see Section 4.3 from our updated draft for very small models).
>
> Regarding how well our paper fits in the datasets and benchmarks track, NeurIPS 2023 explicitly details in the call for papers that this track welcomes “benchmarks on new or existing datasets”.  We benchmark models on a wide variety of existing datasets and present an associated meta-analysis and code that can be used to add subsequent models to the benchmark.  Therefore, our work fits well into the NeurIPS datasets and benchmarks track.  Last year’s NeurIPS datasets and benchmarks tracks included many such papers which studied performance differences between models across existing datasets, for example in [tabular data](https://openreview.net/forum?id=Fp7__phQszn) or to benchmark [OOD detection](https://openreview.net/forum?id=gT6j4_tskUt).
>
> Our focus on existing publicly available checkpoints reflects exactly the kinds of models that are available to practitioners off-the-shelf for deployment.  While we agree with you that different existing backbones are pretrained using different training routines, datasets, and hyperparameter sweeps, we do compare diverse backbones trained on the same datasets and architectures.  Moreover, we already spent roughly 127k GPU hours on our experiments without doing any pretraining ourselves, and pretraining such backbones from scratch ourselves would be prohibitively expensive.
>
> Thank you again for your thoughtful review. We made an effort to address your feedback and would appreciate it if you would consider raising your score in light of our response.  Please let us know if you have additional questions we can address.

---

> > ### Comment · Reviewer_eLJh · 2023-08-29
> >
> > Thank the authors for replying to my concerns with a detailed answer. I also agree that this paper can be quite useful for practitioners, thus I agree with reviewing my rating.

---

> > > ### Author Response · Authors · 2023-08-29
> > > **Thank you for your feedback**
> > >
> > > We thank you again for your feedback, which has strengthened our paper, and for agreeing to revise your rating.

---

### Decision · Program_Chairs · 2023-09-22

**Decision:**

Accept (Poster)

**Comment:**

This paper received a majority of acceptance votes from the reviewers, with a consensus praising its comprehensive study and clear, informative takeaways. One reviewer offered a borderline evaluation due to a lack of independent one-to-one training and insufficient exploration of new combinations, alongside missing object detection and segmentation analysis; this AC concurs that the raised point is valid. Although the authors attempted to address this with some out-of-distribution (OOD) object detection experiments, this AC recommends further experiments on challenging object detection benchmarks and offering more analyses, as suggested by reviewer eLJh. Furthermore, expanding comparisons as advised by the reviewer, would strengthen the paper.

Lastly, this AC advises the authors to revise the paper by addressing the reviewers' outstanding concerns in the final version, encompassing 1) the narrow range of utilized backbones and pretraining methods; 2) the limited scope of tasks and employed datasets, some of which are misaligned due to domain gaps; 3) the absence of a detailed rationale elucidating why a specific backbone demonstrates superior performance. Moreover, please improve the readability of some figures (e.g., specifically address the legends in Figure 1). Overall, this paper merits acceptance.